# Particle Microphysical Parameters and the Complex Refractive Index from 3β + 2α HSRL/Raman Lidar Measurements: Conditions of Accurate Retrieval, Retrieval Uncertainties and Constraints to Suppress the Uncertainties

**Alexei Kolgotin [1], Detlef Müller [2,*] and Anton Romanov [3]**

[1] A.M. Prokhorov General Physics Institute, Moscow 119991, Russia; kolgotin@pic.troitsk.ru
[2] School of Remote Sensing and Information Engineering, Wuhan University, Wuhan 430072, China
[3] Certification Body "Metalcertificate", The National University of Science and Technology, Leninskii av. 4, Moscow 119049, Russia
[*] Correspondence: dgmueller@whu.edu.cn

**Abstract:** We study retrieval methods in regard to their potential to accurately retrieve particle microphysical parameters (PMP) from 3β + 2α HSRL/Raman lidar measurements. PMPs estimated with these methods are number, surface-area and volume concentrations, the effective radius, and complex refractive index of the investigated particle size distribution (PSD). The 3β + 2α optical data are particle backscatter coefficients at 355, 532 and 1064 nm and extinction coefficients at 355 and 532 nm. We present results that are fundamental for our understanding of how uncertainties of the optical data convert into uncertainties of PMPs. PMPs can only be retrieved with preset accuracy if the input optical data are accurate to at least eight significant digits, i.e., $10^{-6}$%. Such measurement accuracy cannot be achieved by currently existing lidar measurement techniques and the fact that atmospheric conditions are not static during lidar observations. Our analysis of the results derived with the novel approach shows that (a) the uncertainty of the retrieved surface-area concentration increases proportionally to the measurement uncertainty of the extinction coefficient at 355 nm, (b) the uncertainty of the effective radius is inversely proportional to the measurement uncertainty of the extinction-related Ångström exponent, (c) the uncertainty of volume concentration is close to the one of the effective radius, and (d) the uncertainty of number concentration is proportional to the inverse of the square value of the uncertainty of the effective radius. The complex refractive index (CRI) cannot be estimated without introducing extra constraints, even if measurement uncertainties of the optical data are as low as 1−3%. We tested constraints and their impact on the solution space, and in how far these constraints could allow us to restrict the retrieval uncertainties. For example, we used information about relative humidity that can be measured with Raman lidar. Relative humidity is an important piece of information that allows for more accurate aerosol typing and thus plays a vital role in any kind of aerosol characterization. The measurement example we used in this study shows that such a constraint can reduce the retrieval uncertainty of single scattering albedo (SSA) to as low as ±0.01−±0.025 (at 532 nm), on the condition that the uncertainty of the input optical data stays below 15%. The results will be used for uncertainty analysis of data products provided by future versions of the Tikhonov Advanced Regularization Algorithm (TiARA). This algorithm has evolved into a standard tool for the derivation of microphysical particle properties from multiwavelength High-Spectral-Resolution Lidar (HSRL)/Raman lidar operated in Europe, East Asia, and the US.

**Keywords:** multiwavelength lidar; atmospheric aerosol optical and microphysical properties; lidar inversion technique

## 1. Introduction

In the past two decades, different types of methods were developed for the retrieval of atmospheric particle microphysical parameters (PMP) including the complex refractive

index (CRI) from optical data taken with lidar [1–6]. In particular, TiARA (version 1.0) has evolved into an algorithm that is now used for the autonomous, i.e., unsupervised, automated near-real-time retrieval of PMPs such as number, surface-area and volume concentration, the effective radius, and the CRI of the investigated particle size distributions [7]. The optical data that are used for the most advanced algorithms require as minimum input particle backscatter coefficients ($\beta$) measured at 355, 532, and 1064 nm, and particle extinction coefficients ($\alpha$) measured at 355 and 532 nm. That set of data naturally requires the use of High-Spectral-Resolution lidar (HSRL) or Raman lidar [8,9] as high-quality optical data, i.e., low measurement uncertainties are needed in order to derive microphysical properties of sufficient accuracy [10].

The use of these retrieval methods requires us overcoming many challenges, i.e.,

1.  The number of input optical data—we denote them as $3\beta + 2\alpha$ set (or simply "3 + 2"), is very limited. Thus, the algorithms need to make optimum use of the information content included in this low number of optical input data.
2.  Measurement errors of the individual optical data points ($\varepsilon_g$) usually are 10% or higher unless we apply unrealistically long data averaging times (in space and time); the parameter $g$ denotes $\alpha$ or $\beta$.
3.  The underlying mathematical problem is ill posed, underdetermined, sensitive to measurement errors, and does not have a unique solution [11].
4.  The identification of the final solution space (a) requires the use of a priori information about particle optical and microphysical properties, (b) forces us to apply mathematical and/or physical constraints to determine the final solution space, and (c) calls for specialized software and data operators who possess wide experience in the specific type of data analysis connected to the problem of dealing with ill-posed inverse problems.

One of the important features of the underlying inverse problem is the particle complex refractive index. This parameter in general is not known. In the ideal case, it should be estimated simultaneously with other PMPs of interest, i.e., the effective radius, and number, surface-area and volume concentrations. The uncertainty of the retrieved CRI usually leads to additional uncertainties of the retrieved PMPs. Therefore, even if the optical data have no measurement uncertainties, which of course is an unrealistic scenario, the methods currently employed do not allow for retrieving the PMPs with preset accuracy.

Another challenge lies in the fact that it is still an open topic to develop methods that allow us to compute in an unambiguous way the uncertainty of the PMPs from the inversion methodologies. One approach involves the use of statistical methods for estimating the properties of uncertainties, or in a broader sense the solution space of uncertainty, as for example mean uncertainty, variance, percentiles, etcetera [7,10]. The analysis of the information content of the measurements (optical input data) is a powerful tool that can be used for the assessment of the level of quality of the retrieved data products, i.e., accuracy, precision; or in more general terms: uncertainty of data products [12,13]. However, such approaches do not provide us with explicit ways of estimating the retrieval uncertainties in a constructive, i.e., mathematically rigorous way. All methods used up to now are qualitative in nature, as for example eigenvalue analysis of the underlying mathematical problem or the assessment of condition numbers of matrices involved in the computations. Aside from this fact, it has become clear in the past two decades from work in, e.g., EARLINET (European Aerosol Research Lidar Network; www.earlinet.org (accessed on 10 April 2023)) and ACTRIS (Aerosol, Clouds and Trace Gases Research Infrastructure; www.actris.eu (accessed on 10 April 2023)) that we do not only need methodologies that allow us to estimate uncertainties of PMPs caused by the inversion methodology (retrieval) method itself. We also need mathematically sound methods that allow us to assess how and to what degree measurement uncertainties contribute to these retrieval uncertainties.

The aim of this study is to develop:

(1)    A reliable, mathematically sound minimization procedure, i.e., a method that allows us to estimate PMPs with preset accuracy and with a preset confidence level, an uncertainty that will not become larger than what can be estimated by this minimization procedure.

(2)    An uncertainty-analysis methodology that can be applied for any method that is used for retrieving PMPs from optical data taken with lidar and other remote sensing instruments, e.g., sun photometer and polarimeter and most importantly: the combination of data collected simultaneously (time and space) with active and passive remote sensors.

(3)    An approach for the identification of (physical and mathematical) constraints that allow us to restrict the (full) solution space obtained from the inversion process so that the final solution can be determined.

In the present study, we focus on the following topics:

1.    Identification of the solution space of the CRI;
2.    Analysis of the solution space structure, convexity and local minima;
3.    Retrieving the PMPs for the values of the CRI identified in point 2;
4.    Uncertainty estimations of the retrieved PMPs in dependence of the values of the measurement uncertainties.
5.    Exploration of the concept of using hygroscopic growth of aerosol pollution under varying relative humidity (RH) conditions for constraining the solution space of the CRI and continuation of the study described in [14].

We are confronted with an (up to now) unquantifiable number of unknown factors that may have significant or less significant impact on the final solution space. There exists a "zoo" of types of aerosol mixtures which in addition constantly changes in the atmosphere (space and time). Any extra information apart from "3 + 2" data is thus not only valuable but vital in providing the best possible solution space of PMPs.

If we knew more about the hygroscopic growth of particles, we could make use of this information for our aerosol characterization in a substantial way. Here, we study if knowledge about hygroscopic properties of aerosol particles can be included in PMP retrieval routines. We are not aware of any online monitoring method that would allow us to constantly adjust latest knowledge on RH growth to our (on-line) data analysis algorithm. However, ACTRIS/EARLINET have a network of Raman lidar stations that either measure or can easily be upgraded to deliver relative humidity and can provide this information in connection to 3 + 2 observations on the global scale with high spatio-temporal resolution.

Section 2 presents the methodology including descriptions of (a) the mathematical problem that needs to be solved, (b) the minimization procedure, and (c) the uncertainty analysis scheme. In Section 3, we describe the simulation procedure and present numerical examples. Section 4 presents the case study of a measurement example. Section 5 summarizes our findings.

## 2. Methodology

### 2.1. Ill-Posed Problem: Choice of Solution Method

Optical particle properties $g(\lambda)$ measured with lidar, i.e., extinction $\alpha(\lambda)$ and backscatter $\beta(\lambda)$ coefficients at wavelength $\lambda$ are related to the unknown particle size distributions $f(r)$ via a Fredholm integral equation of the 1st kind

$$\int_0^\infty K_g(\lambda, m, r) f(r) dr = g(\lambda) \quad g = \alpha, \beta \tag{1}$$

Here, we consider only spherical particles. Therefore, the kernels of the integral equation $K_g(\lambda, m, r)$ are described by the theory of light scattering by spherical particles (Mie-scattering theory) of radius $r$ and the complex refractive index $m = m_R - im_I$ [15].

For naturally occurring particles, particle radius and CRI are approximately limited by the domains

$$r \in [0.01; 20] \, \mu m \qquad m_R \in [1.3; 1.8] \qquad m_I \in [0; 0.1] \qquad (2)$$

In this study, the CRI is assumed to be spectrally independent, though CRIs of naturally occurring particles are wavelength- as well as size-dependent. This assumption obviously is one of the more significant simplifications we still need to apply for practical considerations in our retrieval methods. As we mentioned before, the CRI usually is unknown and therefore also needs to be identified from the inversion of the optical data, aside from the particle size distribution. The PSD defines the concentration of particles per radius interval d$r$ and can be expressed in terms of the distribution of number, surface-area, or volume concentration. The PSD allows us to estimate the bulk values of the PMPs such as the effective radius ($r_{eff}$), number ($n$), surface-area ($s$) and volume ($v$) concentrations [2].

Solving Equation (1) is an ill-posed problem and accompanied with challenges listed in the introduction. The methodology that describes how this ill-posed problem can be solved is given in [11,16,17]. This methodology was adopted in [1–5] and allows us to retrieve the PMPs from optical data taken with multiwavelength HSRL/Raman lidars. The main idea of the methodology includes the inversion of Equation (1) and applying methods of regularization on the solution space. However, inversion with regularization does not guarantee that we obtain stable, physically meaningful solutions of Equation (1). We have to include extra constraints in the retrieval methods so that we subsequently can find the final solution space.

One of the most effective constraints that was developed in the past used a function sample, a model that can properly approximate the investigated PSD. The most popular function samples are uniform, have a logarithmic-normal shape and may consist of a combination of these distributions. These functions are used for the retrievals of PSDs in different applications [6,18–21]. Such an approach, which is denoted as the *direct method*, allows us to reformulate the task of solving Equation (1) in terms of a *parametric* problem rather than a typical inversion problem. In this *direct method*, we fit the unknown parameters that describe the parameters of interest (PSD and CRI) such that the solution of the direct task (1) reproduces the measurements $g(\lambda)$ in the best possible way. Thus, using the model of a uniform or logarithmic-normal distribution for describing the PSDs in the direct methods allows us to stabilize the solution of the ill-posed problem (1) without regularization [11].

An in-depth analysis of the results obtained from the available methods shows that these methods produce the solution spaces with structures that are quite similar to each other. The final solutions converge when we apply the same constraints. In view of this finding, we chose the *direct method* to solve the Equation (1). It represents a key tool suitable for learning more about features of the different algorithms used for the retrieval of PMPs [1–6].

### 2.2. Minimization Procedure

One of the disadvantages of the retrieval methods described in [1–6] is that they use discrete values of real and imaginary parts, i.e., a fixed grid of the CRI. That grid which has a finite resolution on the domain shown in (2) serves as input to Equation (1). No matter what kind of resolution we use for the real and imaginary parts, it is virtually impossible to find a grid resolution (nodes) that coincides with the true value(s) of the CRI we are looking for. In that respect, particularly the fact that naturally occurring particles possess size- and wavelength dependent CRIs exposes one of the main challenges we are confronted with in future research work.

The CRI the way it is used in Equation (1) results in an approximation which adds another layer of uncertainty to the retrieval results of the PMPs. This uncertainty occurs even if measurement uncertainties are zero. Therefore, if we want to improve the quality of the PMPs, i.e., reduce the retrieval uncertainties, we need to modify our retrieval methodologies. We need to find a way that allows us to explicitly retrieve the CRI, which is

already performed in a rather successful fashion for the size parameters of the investigated PSDs [7,22].

In one of our recent publications, we have shown that the CRI $m = m_R - im_I$ can be explicitly retrieved from $3\beta + 2\alpha$ lidar data if the particle size distribution $f(r)$ is known [23]. That approach is based on the minimization of the functional $F_g$

$$F_g^2 = \frac{1}{5} \sum_{g(\lambda_l)} \rho_{g(\lambda_l)}^2(\mathbf{x}) \to \min_{\mathbf{x}}, \qquad g = \alpha, \beta, l = 1, 2, 3 \qquad (3)$$

The functional $F_g$ is expressed as superposition of squares of individual discrepancies

$$\rho_{g(\lambda_l)}(\mathbf{x}) = \frac{|g_{inp}(\lambda_l) - g(\lambda_l, \mathbf{x})|}{g_{inp}(\lambda_l)} \qquad g = \alpha, \beta, l = 1, 2, 3 \qquad (4)$$

where 5 coefficients $g_{inp}(\lambda_l)$ are measured with lidar ($3\beta + 2\alpha$ data) and the 5 coefficients $g(\lambda_l, \mathbf{x})$ are defined by the integral expressions

$$g(\lambda_l, \mathbf{x}) = \int_0^\infty K_g(\lambda_l, x_1 - ix_2, r) f(r) dr \qquad g = \alpha, \beta, l = 1, 2, 3 \qquad (5)$$

Here, the vector $\mathbf{x}$ consists of the 2 elements $\{x_1, x_2\} = \{m_R, m_I\}$. For the implementation of the $F_g^2$-minimization method we used the standard method of cyclic descent [24].

We briefly summarize the main steps of the cyclic descent method for the case of 2 unknown variables. As starting point we set the arbitrary point $\mathbf{x}^0 = \{x_1^0, x_2^0\}$ within the domain (2) and a threshold value $\Delta_F$. In the 1st iteration ($k = 1$), one of the variables, for example $x_1^0$ is fixed and we work out the one-dimensional minimization problem for the 2nd variable, i.e., $x_2$, in the functional $F_g^2$. This problem can be solved, for example, with the golden section method [24] that determines the new value $x_2^1$. Then, in the next step, this variable $x_2^1$ is fixed and we solve the one-dimensional minimization problem for the 1st variable, $x_1$. After the 1st iteration we obtain the new point $\mathbf{x}^1 = \{x_1^1, x_2^1\}$ for which $F_g^{(1)} \le F_g^{(0)}$. The iterations are repeated until we obtain the number $k$ for which the following condition is fulfilled

$$|F_g^{(k)} - F_g^{(k-1)}| \le \Delta_F. \qquad (6)$$

The possibility to retrieve the CRI in the case that the investigated PSD is known agrees with the postulate formulated in [25]. In [25], we postulated the following: the effective radius $r_{eff}$, the lidar ratios $\Lambda(\lambda_l) = \alpha(\lambda_l)/\beta(\lambda_l)$ at $\lambda_1 = 355$ and $\lambda_2 = 532$ nm, and the backscatter- $[\dot{\beta}\left(\frac{\lambda_1}{\lambda_2}\right)]$ and extinction-related $[\dot{\alpha}\left(\frac{\lambda_1}{\lambda_2}\right)]$ Ångström exponents (BAE and EAE, respectively) at the wavelength pair $\lambda_1 = 355$ and $\lambda_2 = 532$ nm uniquely define the CRI of aerosols described by monomodal PSDs.

Let us consider if the problem formulated by Expression (3) can be solved when the PSD is not known. In the case of direct retrieval methods, the monomodal PSD is approximated, for instance, by logarithmic-normal distributions (*LN*) with 3 unknown parameters. The 3 unknown parameters can be, for example, mean radius $r_0$, variance $\sigma$, and number concentration $n$ [6,19]:

$$f(r) = nLN(r_0, \sigma, r) = \frac{n}{(2\pi)^{1/2} r \ln \sigma} \exp\left[-\frac{(\ln r - \ln r_0)^2}{2(\ln \sigma)^2}\right] \qquad (7)$$

We investigate what elements need to be added to the vector $\mathbf{x}$ so that we may use the minimization procedure (3) to retrieve both, the CRI $m = m_R - im_I$ and the PSD $f(r)$. In the first step, we refine the postulate formulated in [25].

Firstly, one can show that four intensive parameters $\Lambda(355)$, $\Lambda(532)$, $\dot{\beta}\left(\frac{355}{532}\right)$ and $\dot{\alpha}\left(\frac{355}{532}\right)$ measured with lidar are not linearly independent. Either one of these 4 intensive parameters (IP) can be expressed by the 3 other parameters [26,27]. In order to keep the 4 pieces of

independent information, we can replace the parameter $\Lambda(532)$ with $\dot{\beta}\left(\frac{\lambda_2}{\lambda_3}\right)$ at the pair of wavelengths $\lambda_2 = 532$ and $\lambda_3 = 1064$ nm [13]. For convenience we will only use extinction and backscatter ratios instead of their respective exponents:

$$A\left(\frac{\lambda_l}{\lambda_{l+1}}\right) = \frac{\alpha(\lambda_{l+1})}{\alpha(\lambda_l)} = \left(\frac{\lambda_l}{\lambda_{l+1}}\right)^{\dot{\alpha}(\lambda_l/\lambda_{l+1})} \quad B\left(\frac{\lambda_l}{\lambda_{l+1}}\right) = \frac{\beta(\lambda_{l+1})}{\beta(\lambda_l)} = \left(\frac{\lambda_l}{\lambda_{l+1}}\right)^{\dot{\beta}(\lambda_l/\lambda_{l+1})} \quad l = 1, 2 \tag{8}$$

Secondly, the effective radius is a compound parameter which can be uniquely represented by the 2 free/simple parameters $r_0$ and $\sigma$. Thus, the 4 parameters

$$\mathbf{x} = \{x_i\} = \{m_R, m_I, r_0, \sigma\} \qquad i = 1, \dots, 4 \tag{9}$$

drive the 4 IPs

$$\mathbf{G} = \{G_j\} = \left\{ A\left(\frac{355}{532}\right), B\left(\frac{355}{532}\right), B\left(\frac{532}{1064}\right), \Lambda(355) \right\} \qquad j = 1, \dots, 4 \tag{10}$$

From the theoretical point of view, if interdependencies between four PMPs and four IPs are correlated one-to-one we deal with a closed mathematical system. That means we can find a solution to the task of identifying 4 elements (9) from 4 IPs (10). In this case, the refined version of the postulate is formulated as follows: mean radius $r_0$, variation $\sigma$, and CRI $m = m_R - im_I$ of aerosols described by monomodal PSDs uniquely define:

-　　The lidar ratio $\Lambda(\lambda_l)$ at one of the wavelengths, i.e., either at $\lambda_1 = 355$ or $\lambda_2 = 532$ nm,
-　　The backscatter- $[\dot{\beta}\left(\frac{\lambda_1}{\lambda_2}\right)]$ and extinction-related $[\dot{\alpha}\left(\frac{\lambda_1}{\lambda_2}\right)]$ Ångström exponents at the wavelength pair $\lambda_1 = 355$ and $\lambda_2 = 532$ nm, and
-　　The backscatter-related Ångström exponent $[\dot{\beta}\left(\frac{\lambda_2}{\lambda_3}\right)]$ at the wavelength pair $\lambda_2 = 532$ and $\lambda_3 = 1064$ nm.

The refined postulate allows us to reformulate the minimization procedure (3) in terms of IPs (10) and the vector **x** that contains the elements (9) so that the functional

$$F_G^2 = \frac{1}{4} \sum_{j=1}^{4} \rho_{G_j}^2(\mathbf{x}) \to \min_{\mathbf{x}} \tag{11}$$

is minimized. The functional is expressed through the individual discrepancies

$$\rho_{G_j}(\mathbf{x}) = \frac{\left|G_{\text{inp},j} - G_j(\mathbf{x})\right|}{G_{\text{inp},j}} \qquad j = 1, \dots, 4 \tag{12}$$

where the elements $G_{\text{inp},j}$ and $G_j(\mathbf{x})$ are defined on the basis of coefficients $g_{\text{inp}}(\lambda_l)$ and $g(\lambda_l, \mathbf{x})$, respectively [see, for example, Equation (8)]. In turn, the 5 coefficients $g_{\text{inp}}(\lambda_l)$ are known from the measurements whereas the 5 coefficients $g(\lambda_l, \mathbf{x})$ are computed by the equation

$$g(\lambda_l, \mathbf{x}) = \int_0^\infty K_g(\lambda_l, x_1 - ix_2, r) LN(x_3, x_4, r) dr \qquad g = a, b, l = 1, 2, 3 \tag{13}$$

Obviously, the elements of vector **G** do not depend on the concentration $n$ and therefore we are left with 4 unknown variables in Equation (13).

We can modify the minimization procedure introduced in [23], i.e., we add another pair of unknown parameters $\{x_3, x_4\} = \{r_0, \sigma\}$. In the case of monomodal PSDs, these parameters allow us to approximate PSDs comparably accurately by logarithmic-normal distributions (7). Approximation errors are in fact absent if the unknown PSD is described by a log-normal distribution as well. Obviously, we can use the cyclic descent method described above for the case in which the vector **x** consists of 4 elements. In this case, we consider the one-dimensional minimization problem for one of the 4 variables. The other 3 variables remain fixed in the functional $F_G$ in each iteration step $k$. We note that the

number concentration can be obtained from the equation $n = g_{\mathrm{inp}}(\lambda_l)/g(\lambda_l, \mathbf{x})$. In Section 3, we will investigate in more detail how the minimization procedure can be used for solving Equation (1) and what uncertainties it produces for the unknown parameters.

*2.3. Uncertainties of Retrieved Microphysical Parameters*

In this section, we present the results of the uncertainty analysis, i.e., we investigate the magnitudes of retrieval uncertainties versus measurement uncertainties for $r_{\mathrm{eff}}$, $n$, $s$ and $v$ of the PMPs. We use the absolute ($\Delta_p$) or relative ($\varepsilon_p$) deviation between the parameters $p^\varepsilon$ and $p$,

$$\Delta_p = p^\varepsilon - p \qquad \varepsilon_p = p^\varepsilon/p - 1 \tag{14}$$

respectively. In our investigation, we make the following assumptions:

1.  The CRI is known for the moment or we have an estimation of the CRI at a given value $m$.
2.  The PSD is monomodal and defined for the accumulation mode. Without loss of generality we write the conditions for mean radius and EAE of the accumulation mode as

$$r_0 \geq 0.075 \ \mu\mathrm{m} \qquad \dot{\alpha}\left(\frac{355}{532}\right) \in (0.2; \ 2.0). \tag{15}$$

3.  Particle backscatter coefficients measured with lidar are more accurate than particle extinction coefficients. Therefore, if we use for example $\varepsilon_\alpha = 15\%$ uncertainty for the extinction coefficient, the uncertainty of the backscatter coefficient is less, i.e., $|\varepsilon_\beta| \leq |\varepsilon_\alpha| = 15\%$.
4.  Backscatter coefficients are sensitive to properties of the PMPs including their CRIs. In contrast, extinction coefficients are sensitive to size and integral properties of the PMPs, only. In view of assumption # 3, we can focus on investigating the uncertainties of the PMP retrieval versus uncertainties of the measured extinction coefficients $\varepsilon_\alpha$ [28].
5.  We also consider the most extreme uncertainty scenarios, i.e., the uncertainties of the two extinction coefficients have opposite sign

$$\varepsilon_{\alpha(355)} = -\varepsilon_{\alpha(532)} \tag{16}$$

For example, if the uncertainty of $\alpha(355)$ is $\varepsilon_{\alpha(355)} = +15\%$ then the error of $\alpha(532)$ is $\varepsilon_{\alpha(532)} = -15\%$.

The uncertainty analysis scheme developed in this paper uses the correlation relationships between the PMPs and the optical data, see our recent study [25]. We also use the *etalon* solution space. We store this *etalon* solution space in our reference look-up table (RLUT) that contains the synthetic optical data together with the PMPs. The RLUT contains 408 different CRIs and 846 different PSDs. The PSDs are described by different logarithmic-normal distributions:

$$m_{\mathrm{R}} \in [1.3;1.7]; \quad m_{\mathrm{I}} \in [0;0.05]; \quad r_0 \in [0.015;6.3] \ \mu\mathrm{m}; \quad \sigma \in [1.35;2.55]; \quad n = 1 \ \mathrm{cm}^{-3} \tag{17}$$

The current version of our RLUT consists of 345,168 optical datasets of $3\beta + 2\alpha$. We stress that the extended RLUT and the assumptions 1 and 2 allow us to estimate in a more precise way the coefficients of the regression equations that describe the correlation relationships.

We start the uncertainty analysis by estimating the retrieval uncertainty of surface-area concentration. We showed in [25] that surface-area concentration and extinction coefficient at 355 nm are linearly correlated. The correlation coefficient $R$ is comparably high. The following relationship can be used:

$$s = a_s\alpha(355) \pm 20\% \tag{18}$$

For a mixture of small and large particles, i.e., PSDs represented by fine and coarse modes, this equation uses the regression coefficient $a_s = 1.6$.

Now we consider the PSDs that contain the accumulation mode only. We collected all entries in our RLUT for this case [see conditions (15)] and found the regression equations at different CRIs (see Figure 1a). We see from Figure 1a that

- Surface-area concentration is strongly correlated with the extinction coefficients at 355 nm. We find a high correlation coefficient that exceeds $R > 0.99$ and
- The regression coefficient $a_s$ increases from 1 at $m = 1.7$-$i0 \ldots 0.05$ to 1.19 at $m = 1.325$-$i0 \ldots 0.0075$.

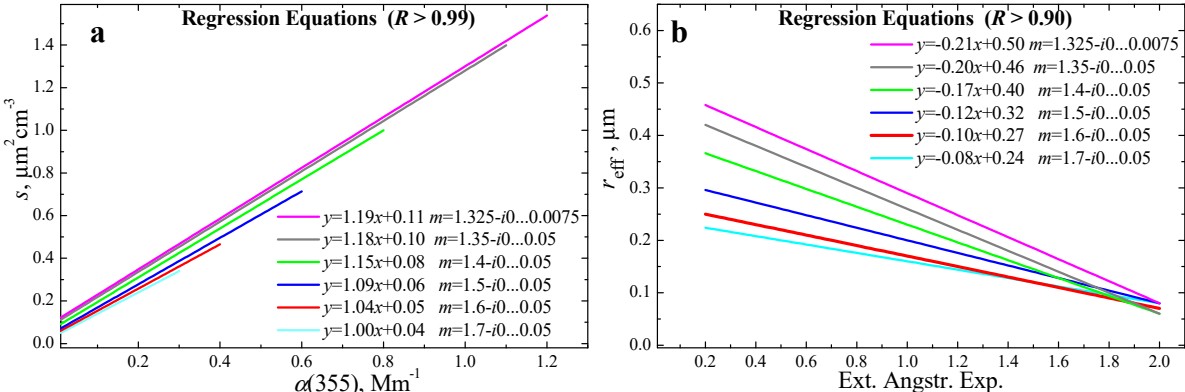

**Figure 1.** Graphical representations of the regression equations that describe the correlations "surface-area concentration vs. extinction coefficient at 355 nm" (**a**) and "effective radius vs. EAE" (**b**) for different CRIs (see legends) and accumulation modes of PSDs [see conditions (15)] in the RLUT. The minimum values of the correlations coefficients among all regression equations are $R = 0.99$ and $R = 0.90$, respectively.

This result means that the retrieval uncertainty of the surface-area concentration $\varepsilon_s$ and the measurement uncertainty of the extinction coefficient $\varepsilon_{\alpha(355)}$ are almost proportional to each other (see Figure 2a), i.e.,

$$\varepsilon_s \approx \varepsilon_{\alpha(355)} \tag{19}$$

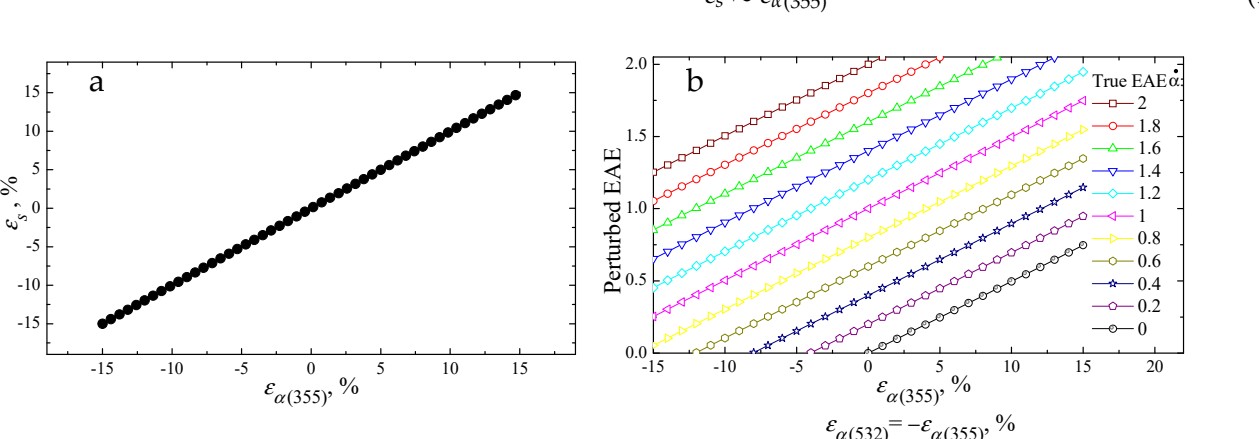

**Figure 2.** (**a**) Retrieval uncertainty of surface-area concentration versus measurement error of extinction coefficient at 355 nm. (**b**) Perturbed EAE according to the true values $\dot{\alpha} = 0, 0.2, \ldots, 2$ (see legend) versus measurement uncertainties of the extinction coefficients $\varepsilon_\alpha$.

We apply the same approach to estimate the retrieval uncertainty for the effective radius. In our extended RLUT, we find that $r_{\text{eff}} \in (0.09; 0.5)$ µm fulfills the conditions (15). We have shown in [25] that the effective radius and EAE are linearly correlated, i.e.,

$$r_{\text{eff}} = a_r \, \dot{\alpha} \, + \, b_r \tag{20}$$

The correlation coefficient is comparably high.

Figure 1b shows the regression equations that describe the correlation relationships between $r_{\mathrm{eff}}$ and $\dot{\alpha}$ at different CRIs. The regression coefficients $|a_r|$ and $b_r$ increase from 0.08 and 0.24 at $m = 1.7$-$i0 \ldots 0.05$ to 0.21 and 0.5 at $m = 1.325$-$i0 \ldots 0.0075$, respectively. The correlation coefficient is $R = 0.9$ and above at the fixed real parts of CRI as shown in Figure 1b.

The high level of the correlation coefficient means that the EAEs contain significant information about the magnitude of the effective radius. Therefore, variations of the EAE near their true value can be used to predict the respective variations of the effective radius, simply by using the regression equations (see legend in Figure 1b). Thus, we can analyze what level of perturbations of the EAEs is generated by the measurement uncertainties (16). In the case of the uncertainty scenarios (16) that we are considering in this study, the perturbed EAE is

$$\dot{\alpha}^{\varepsilon} = \ln^{-1}\frac{355}{532}\ln\frac{\alpha^{\varepsilon}(532)}{\alpha^{\varepsilon}(355)} = \ln^{-1}\frac{355}{532}\ln\frac{\alpha(532)\left(1 - \varepsilon_{\alpha(355)}\right)}{\alpha(355)\left(1 + \varepsilon_{\alpha(355)}\right)} \approx \dot{\alpha} + 5\varepsilon_{\alpha(355)} \qquad (21)$$

Figure 2b shows the perturbed EAEs near the vicinity of the true values, i.e., $\dot{\alpha} = 0$, $0.2, \ldots, 2$ (see legend) versus the measurement uncertainties of the extinction coefficients. For example, if the true EAE is $\dot{\alpha} = 0$, an uncertainty of $+15\%$ of the extinction coefficients produces a perturbation of $\dot{\alpha}^{\varepsilon} \approx 0.75$ of the EAE (see black line and circle, trajectory in the lower right portion of the plot).

In view of the nearly inverse proportionality of $r_{\mathrm{eff}}$ and $\dot{\alpha}$, the retrieval uncertainty of the effective radius is also inversely proportional to the measurement uncertainty of the EAE for fixed values of the real part of the CRI. We can find the retrieval uncertainty of the effective radius in dependence of each perturbed value of the EAE by investigating the vicinities of the different values of $\dot{\alpha} = 0, 0.2, \ldots$ or 2 (see Figure 3a). For example, the regression equation at $m = 1.5$-$i0 \ldots 0.05$ is $y = -0.12x + 0.32$ (see legend in Figure 1b), which means:

$$r_{\mathrm{eff}} \approx -0.12\dot{\alpha} + 0.32 \qquad (22)$$

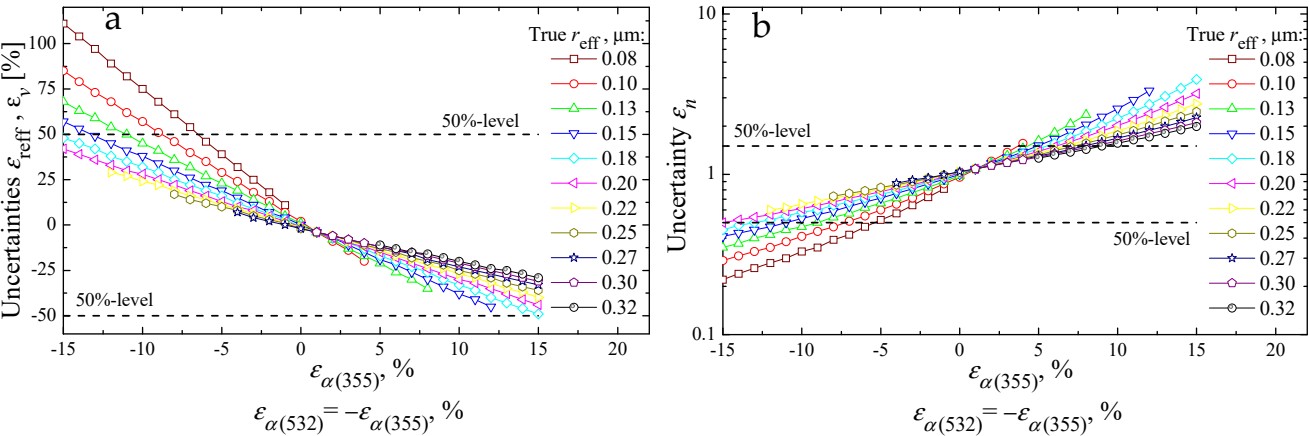

**Figure 3.** (**a**) Retrieval uncertainty of the effective radius $\varepsilon_{\mathrm{reff}}$ (**a**) and number concentration $\varepsilon_n$ (**b**) vs. measurement uncertainties of the extinction coefficients at $m = 1.5$-$i0 \ldots 0.05$. The trajectory of the uncertainty, i.e., uncertainty curve for each true effective radius $0.08, 0.10, \ldots, 0.32$ μm (see legend) is determined by the variation of the respective EAE, see Figure 2b, and the regression equation $y = -0.12x + 0.32$ at $m = 1.5$-$i0 \ldots 0.05$, see Figure 1b. The trajectories of the uncertainty $\varepsilon_v$ for volume concentration and the effective radius $\varepsilon_{\mathrm{reff}}$ are close to each other. The trajectories of the uncertainty of number concentration are determined by the respective trajectories $\varepsilon_{\mathrm{reff}}$ and Equation (30). The dashed lines show the $\pm 50\%$ retrieval uncertainties.

The (perturbed) value of the effective radius can be estimated as

$$r_{\text{eff}}{}^{\varepsilon} \approx -0.12\dot{\alpha}^{\varepsilon} + 0.32 \tag{23}$$

If the true value of the EAE is $\dot{\alpha} = 0$, an uncertainty of +15% of the extinction coefficients, as it was estimated above, produces a perturbation of $\dot{\alpha}^{\varepsilon} \approx 0.75$ of the EAE. This perturbation translates into a value of $r_{\text{eff}}{}^{\varepsilon} \approx 0.23$ µm. The respective retrieval uncertainty is (see black line and circles of the top-right trajectory in Figure 3a):

$$\varepsilon_{r_{\text{eff}}} = r_{\text{eff}}{}^{\varepsilon}/r_{\text{eff}} - 1 \approx 0.23/0.32 - 1 \approx -28\%. \tag{24}$$

We note that in the case of monomodal PSDs the case of $-15\%$ uncertainty of the extinction coefficient at 355 nm leads to an overestimation of +100% of the effective radius (see brown line and squares; top-left trajectory). However, the $r_{\text{eff}}$-retrieval uncertainty is less than 50% for particles with the effective radius exceeding 0.15 µm if measurement uncertainties are less than 15%.

In view of Equations (20) and (21), we can write the equation that describes the uncertainty of the retrieved $r_{\text{eff}}$ in the general case in terms of the measurement uncertainties of the extinction coefficient

$$\varepsilon_{r_{\text{eff}}} = r_{\text{eff}}{}^{\varepsilon}/r_{\text{eff}} - 1 \approx \frac{5\varepsilon_{\alpha(355)}}{\dot{\alpha} + b_r/a_r} \tag{25}$$

The ratio of the regression coefficients fulfils the condition $b_r/a_r \in [-3.0; -2.3]$. This condition range is determined by the 6 real parts presented in Figure 1b. We find a value of $b_r/a_r = -3.0$ at $m = 1.7\text{-}i0 \ldots 0.05$. Thus, we can use the estimation

$$\varepsilon_{r_{\text{eff}}} = -(1.8\ldots16.7)\varepsilon_{\alpha(355)} \tag{26}$$

for the conditions (15) that are applied to EAE. The factor 1.8 corresponds to the highest real part $m_R = 1.7$ and the factor 16.7 to the lowest real part $m_R = 1.35$.

The retrieval uncertainty of the volume concentration $\varepsilon_v$ is determined by the retrieval uncertainties of the surface-area concentration and the effective radius because of the following definition

$$v = s\, r_{\text{eff}}/3 \tag{27}$$

If we take into account the definition for the relative deviation, see (14) and Equation (19), the retrieval uncertainty of the volume concentration is expressed as

$$\varepsilon_v = s^{\varepsilon}\, r_{\text{eff}}{}^{\varepsilon}/s\, r_{\text{eff}} - 1 \approx \varepsilon_{r_{\text{eff}}} + \varepsilon_{\alpha(355)} \tag{28}$$

Here, we neglected the third summand $\varepsilon_{r_{\text{eff}}}\varepsilon_{\alpha}$ because it is infinitesimal in second order. In view of Equation (25), we conclude that

$$\varepsilon_v \leq \varepsilon_{r_{\text{eff}}} \tag{29}$$

because of the compensation effect. Therefore, the element $\varepsilon_{\alpha(355)}$ can be omitted in Equation (27) as well. It means that we consider the worst-case scenario with regard to the magnitude of the retrieval uncertainty $\varepsilon_v$.

Finally, we find the retrieval uncertainty of the number concentration $\varepsilon_n$. We have shown in our study [25] that the following correlation holds true:

$$s \sim n\, r_{\text{eff}}{}^2 \tag{30}$$

In view of Equation (18), the product ($n\, r_{\text{eff}}{}^2$) is approximately constant for fixed extinction coefficient. This property allows us to correlate the retrieval uncertainties of number concentration and the effective radius, i.e., the retrieval uncertainty of number

concentration $\varepsilon_n$ is determined by the square of the $r_{\text{eff}}$-retrieval uncertainty. Since in the case of number concentration the retrieval uncertainty is much larger compared to the other PMP uncertainties (see above), we write this uncertainty as ratio $n^\varepsilon/n$ rather than using the % uncertainty. We obtain

$$n^\varepsilon/n \approx [r_{\text{eff}}/r_{\text{eff}}{}^\varepsilon]^2 = [1 + \varepsilon_{r_{\text{eff}}}]^{-2} \tag{31}$$

Figure 3b shows the different possibilities of the $n$ uncertainty trajectories at $m = 1.5$-$i0$ ... 0.05 if the conditions (15) hold true. We see that a measurement uncertainty as low as +3% to +5% of $\alpha(355)$ may produce a 50% overestimation of number concentration. This result agrees with results we obtained over many years of algorithm development and simulation work. We can once more confirm that number concentration is the most unstable parameter in the retrieval process.

The Expressions (19), (24), (28) and (30) are at the heart of the uncertainty analysis we use for estimating the retrieval uncertainties of $s$, $r_{\text{eff}}$, $v$ and $n$. In the case of surface-area concentration, its uncertainty $\varepsilon_s$ is determined by the uncertainty $\varepsilon_{\alpha(355)}$. As for the PMPs $r_{\text{eff}}$, $v$ and $n$ their uncertainties mainly depend on the uncertainty of the EAE. This uncertainty in turn depends on the accuracy of the measurements of the extinction coefficients.

This uncertainty analysis scheme can also be developed separately for Aitken and coarse mode particles and their PSDs, and for any combinations of Aitken, accumulation and coarse modes of PSDs, respectively. In view of the fact that we considered particles of the accumulation mode, which are the optically most-active ones in the spectral range from 355 to 1064 nm we can conclude that the PMP retrieval uncertainties expressed by (19), (24), (28) and (30) are the best (i.e., minimal) possible values we may expect from $3\beta + 2\alpha$ measurements (see Figure 4).

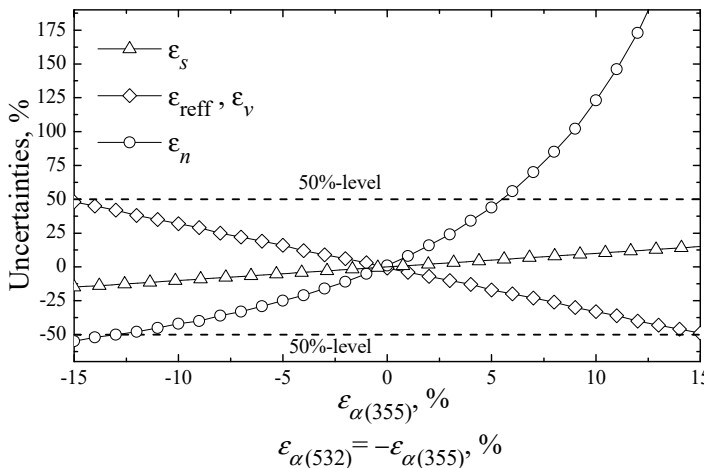

**Figure 4.** Idealized sketch of all retrieval uncertainties: the effective radius (diamond), number (circle), surface-area (triangle) and volume (diamond) concentrations vs. measurement uncertainties of the extinction coefficients $\varepsilon_{\alpha(355)}$ and $\varepsilon_{\alpha(532)}$.

## 3. Simulation

### 3.1. Analysis of Solution Space

We show a numerical example that we used to test our theoretical results. We demonstrate how the minimization procedure and uncertainty analysis work in practice. We used a set of synthetic optical data $3\beta_{\text{inp}} + 2\alpha_{\text{inp}}$. We computed these optical data from Equation (1). We used as input the CRI $m = 1.5$-$i0.001$ and a logarithmic-normal distribution (7) with mean radius $r_0 = 0.22$ μm, variance $\sigma = 1.5$ and number concentration $n = 1$ cm$^{-3}$ (see black curve in Figure 5a).

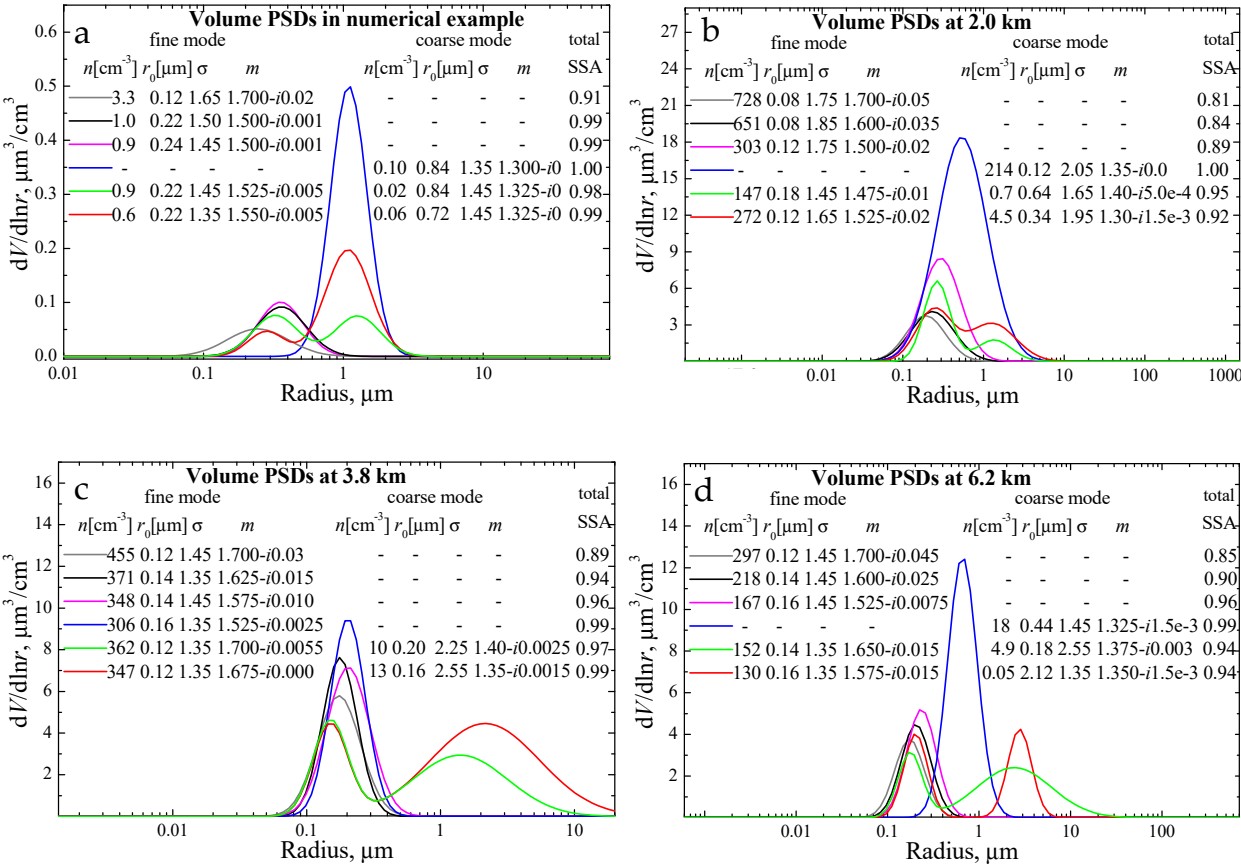

**Figure 5.** Volume PSDs, CRIs and SSA at 532 nm (see respective parameters $n$, $r_0$, $\sigma$, $m$ and SSA in the legends) retrieved from the optical datasets $3\beta + 2\alpha$ used in the numerical example (**a**) and collected during lidar measurements on 22 July 2004 at heights 2.0 km (**b**), 3.8 km (**c**) and 6.2 km (**d**) [see details in Section 4]. The discrepancies (32) for the retrieved solutions are $\rho \in [4\%; 5\%]$ (**a**), $\rho \in [1\%; 4\%]$ (**b**), $\rho \in [11\%; 15\%]$ (**c**) and $\rho \in [15\%; 26\%]$ (**d**) for monomodal PSDs and $\rho < 3\%$ for all bimodal PSDs.

We carried out the proximate analysis (PA) we introduced in [19]. PA allows us to collect from the RLUT those solutions that approximately reproduce the $3\beta + 2\alpha$ set, i.e., we pick all those solutions that produce optical data that are similar compared to the input set of $3\beta_{\text{inp}} + 2\alpha_{\text{inp}}$. We used a discrepancy $\rho$ that did not exceed 5%:

$$\rho = \frac{1}{5}\sum_{g(\lambda_l)}\rho_{g(\lambda_i)} = \frac{1}{5}\sum_{g(\lambda_l)}\frac{|g_{\text{inp}}(\lambda_l) - g(\lambda_l)|}{g_{\text{inp}}(\lambda_l)} \quad g = \alpha, \beta, l = 1, 2, 3 \quad (32)$$

The results of the PA are shown in Figure 5a. We see that the individual solutions differ significantly from each other. Number concentration and mean radius vary from $n = 3.25$ cm$^{-3}$ and $r_0 = 0.115$ µm (gray) to $n = 0.1$ cm$^{-3}$ and $r_0 = 0.835$ µm (blue), respectively. The CRIs of these solutions vary from 1.7-$i$0.02 to 1.3-$i$0, respectively. That range of numbers covers the complete domain of the real part (17) considered in the RLUT. Furthermore, the fixed optical dataset we used for the test can be reproduced by both mono- (see gray, black, pink and blue curves) and bimodal (see green and red curves) PSDs.

We find important patterns from these results:

1.  The larger the (mean) particle size (the lower number concentration), the lower are the real and imaginary parts of the CRI, and conversely.
2.  One and the same optical dataset can be produced by bimodal and monomodal PSDs. In these cases, the CRIs of monomodal PSDs (see black and pink solutions in Figure 5a) are intermediate between the CRIs of the fine and coarse modes of the bimodal PSDs,

respectively (see green and red solutions in Figure 5a). However, single scattering albedo (SSA) of the monomodal (0.99) and bimodal PSDs (0.98–0.99) at 532 nm do not differ significantly.

3.  The number of individual bimodal PSDs retrieved with discrepancy $\rho \leq 5\%$ (Figure 5a shows only 2 of the bimodal PSDs) is larger than the number of individual monomodal PSDs retrieved within the same discrepancy interval (Figure 5a shows only 3 examples of monomodal PSDs). Furthermore, the discrepancies of the bimodal PSD are less than 2% in spite of the fact that we consider for this numerical example optical data that are produced by a monomodal PSD (see Table 1).

4.  The numerical example shown in Figure 5a is typical (representative) for any other (arbitrary) optical dataset $3\beta + 2\alpha$. We stress the fact that the non-uniqueness of the solutions that follow from solving Equation (1) is caused not only by the uncertainty of the optical input data but also by the high degree of freedom the $3\beta + 2\alpha$ datasets permit. We presented and discussed in previous publications examples that cause this high degree of freedom, as for example wavelength-dependent CRIs, shape of PSDs, and the fact that any logarithmic-normal PSD has an analogue solution given by the sum of 2 other logarithmic-normal PSDs. Furthermore, the degree of freedom is much larger for bimodal PSDs. We need to keep this fact in mind when we analyze results retrieved, for instance, with passive remote sensors. In that case, we usually obtain bimodal PSDs [20,29,30].

**Table 1.** Retrieval results and their uncertainties (sub- and superscripts). For details, see Figure 5 and Section 4. The uncertainties expressed in % are estimated with the assumption that (a) the measurement uncertainty is 15% and (b) the PSDs are monomodal. The uncertainties which are expressed in terms of parameter dimensions include the retrieval results of bimodal PSDs. The parameters $\rho_{\text{mono}}$ and $\rho_{\text{bi}}$ denote the discrepancies (31) of monomodal and bimodal PSDs, respectively, that are included in the solution space (see Figure 5).

| Parameter | Numerical Example | Measurements | | |
|---|---|---|---|---|
| | | at 2.0 km | at 3.8 km | at 6.2 km |
| $\rho_{\text{mono}}$, % | 4–5 | 1–4 | 11–15 | 15–26 |
| $\rho_{\text{bi}}$, % | 0.4–1.6 | 0.3–0.4 | 0.2–2.8 | 0.3–2.4 |
| $s$, $\mu\text{m}^2\text{cm}^{-3}$ | $0.85^{+15\%}_{-15\%}$ | $98^{+15\%}_{-15\%}$ | $104^{+15\%}_{-15\%}$ | $66^{+15\%}_{-15\%}$ |
| $r_{\text{eff}}$, $\mu$m | $0.33^{+0\%}_{-28\%}$ | $0.43^{+37\%}_{-37\%}$ | $0.20^{+0.25}_{-40\%}$ | $0.22^{+0.24}_{-38\%}$ |
| $v$, $\mu\text{m}^3\text{cm}^{-3}$ | $0.094^{+0\%}_{-28\%}$ | $14^{+37\%}_{-6}$ | $7^{+7}_{-40\%}$ | $5^{+3}_{-38\%}$ |
| $n$, $\text{cm}^{-3}$ | $1^{+93\%}_{-0\%}$ | $234^{+150\%}_{-47\%}$ | $286^{+188\%}_{-49\%}$ | $144^{+160\%}_{-48\%}$ |
| SSA (532) | $0.99^{+0.01}_{-0.08}$ | $0.975^{+0.025}_{-0.025}$ | $0.975^{+0.015}_{-0.015}$ | $0.95^{+0.01}_{-0.01}$ |

In summary, the solutions we obtain, i.e., PSDs and CRIs are nearly arbitrary but still reproduce the input optical data to sufficiently high accuracy. We also obtain SSA which varies between 0.91 and 1. Such a range of numbers does not have any practical value (see sub- and superscripts of the numerical example in Table 1) and can make measurements of $3\beta + 2\alpha$ data useless. The following questions inevitably come up: (1) *why do the available retrieval methods work* and (2) *why are the results obtained with these methods consistent with results of other measurement techniques?*

The available methods deliver reasonable results because they use a priori information about particle properties and extra constraints that are applied to the initial solution spaces we obtain for different input optical datasets. We can illustrate how a priori/constraint information works for the case of our numerical example. For instance, let us assume that particles are small, as may be the case for fresh anthropogenic pollution, by using knowledge from, e.g., aerosol typing methods. These methods have been developed for HSRL-1 and HSRL-2 [31] and are currently in the development stage in the framework of

ACTRIS [32]. In that case, we can remove the radius domain of the search space above $r = 1$ µm as we know from plenty of literature values that particles of fresh some unlikely possess radii above 1 µm. In this case, the retrieval results will exclude the blue, red and green PSDs from the solution space shown in Figure 5a. If we add extra constraints to the CRI, as for example a limited search space $m_R < 1.7$, the gray solution in Figure 5a will be eliminated in the final solution space, too.

In the next step, we show in what way we can insert requirements for optical data uncertainties that allow us to find accurate and unique solutions.

*3.2. Use of the Minimization Procedure*

The first step of using the minimization procedure consists of redefining the domains of the elements of vector **x**. We use the domain $D$ from our RLUT, i.e.,

$$
\begin{array}{ll}
x_1 = m_R \in [1.3; 1.7]; & x_2 = m_I \in [0; 0.05]; \\
x_3 = r_0 \in [0.015; 6.3] \text{ µm}; & x_4 = \sigma \in [1.35; 2.55];
\end{array}
\tag{33}
$$

There are a few local minima (1–3) on this domain. The magnitude of the local minima covers 5 orders of magnitude

$$
F_g{}^{\min} \in (10^{-4}\%; 10^1\%)
\tag{34}
$$

The input $g(\lambda_l)$, i.e., the optical data need to be accurate to 6 significant digitals, which is of course an unrealistic requirement for the case of lidar observations. Even worse, if we aim at distinguishing the global minima that coincide with the true vector $\mathbf{x}^{\text{true}} = \mathbf{x}^{\text{opt}}$ the input $g(\lambda_l)$ must include even more than 6 digits. We need optical input data that are accurate to 8 digits, which translates into an accuracy of at least $10^{-6}\%$.

The method of cyclic descent that is used in the minimization procedure does not permit for identifying the global minimum if the functional $F_G^2$ in (11) is not convex and has more than 1 minimum. Unfortunately, the $F_G^2$ in our case has a ravine-like, ragged structure and, as we mentioned before, a few local minima. Therefore, we also need to identify all local minima on the domain $D$. We can achieve this task by splitting the domain into parts, i.e., into subdomains (or cells) that contain only one local minimum. Each subdomain can be considered as a space with convex structure.

All minima on the domain $D$ can be identified if we use an important property of the functionals $F_g^2$ and $F_G^2$. We showed in previous publications (see details in [23,25,33]) that the minima of $F_g^2$ and $F_G^2$ are distributed along a "canyon" which can easily be found, as for example with our PA method [19]. In view of this property of $F_g^2$ and $F_{G'}^2$ we scan the domain $D$ with PA first and then we use the minimization procedure only in the cell $D_{ts\mu\nu} \subseteq D$ that lies inside that "canyon". The cell $D_{ts\mu\nu}$ is determined by the local domain

$$
[m_R{}^t; m_R{}^{t+1}] \qquad [m_I{}^s; m_I{}^{s+1}] \qquad [r_0{}^\mu; r_0{}^{\mu+1}] \qquad [\sigma^\nu; \sigma^{\nu+1}]
\tag{35}
$$

where $m_R^t$, $m_R^{t+1}$, $m_I^s$, $m_I^{s+1}$, $r_0^\mu$, $r_0^{\mu+1}$, $\sigma^\nu$, $\sigma^{\nu+1}$ are RLUT elements. For example, the superscripts $t$ and $t + 1$ mean that the respective elements $m_R$ are adjacent to each other in the RLUT.

The minimization procedure uses the preset threshold $\Delta_F$ as input. In view of the magnitude of the global minimum, i.e., $10^{-6}\%$, we can predefine the value $\Delta_F \sim 10^{-8}$.

Figure 6 shows the performance of this approach for the optical dataset we computed for the numerical example presented in Section 3.1. In the minimization procedure, we use the optical datasets on the condition of a preliminary accuracy of 8 (Figure 6a), 3 (Figure 6b) and 2 (Figure 6c) significant digits. These digits reflect an accuracy of $10^{-6}\%$, $10^{-1}\%$, and $10^0\%$, respectively.

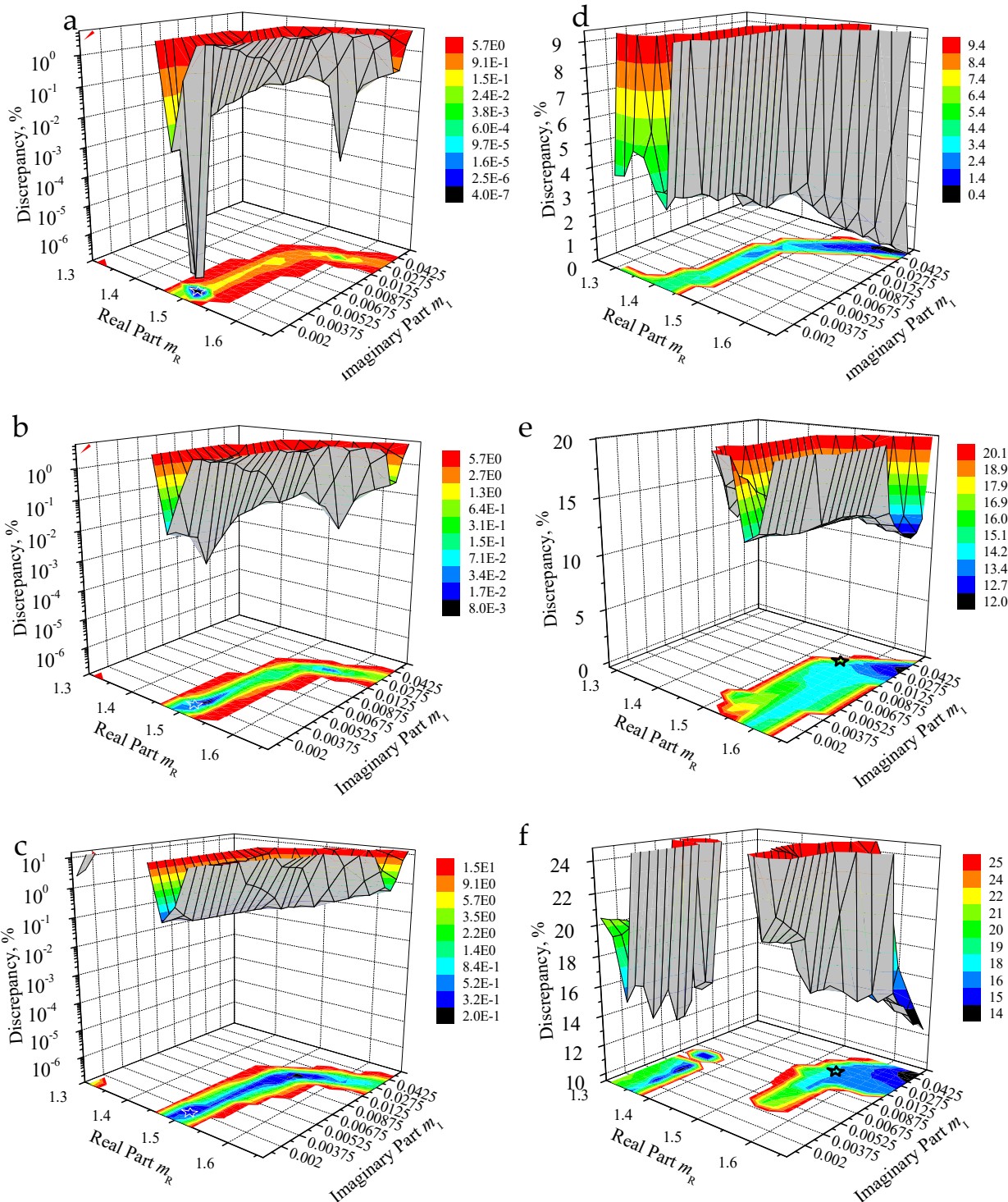

**Figure 6.** Use of minimization procedure in the cases of known (true) optical data (**a**), and optical data that are approximated to an accuracy of 0.1% (**b**) and 1% (**c**). Additionally, shown are results obtained from the lidar data taken on 22 July 2004 at heights 2.0 km (**d**), 3.8 km (**e**) and 6.2 km (**f**) [see details in Section 4]. The true values of CRI, mean radius, variance, and number concentration in the cases (a–c) are $m = 1.5-i0.001$, $r_0 = 0.22$ μm, $\sigma = 1.5$, and $n = 1$ cm$^{-3}$, respectively The stars on the $(m_R, m_I)$ plane show the true solution (white) and the result of the in situ observations (black), i.e., $m = 1.5-i0.001$ and $m \approx 1.55-i0.02$ (see details in Section 4), respectively. Gray color means a rear of surface.

Our analysis of the solutions for this example shows that:

(a)　in the 1st case, there are 3 minima:

| Kind of Minimum | $F_g$ [%] | $m$ | $r_0$ [μm] | $\sigma$ | $n$ [cm$^{-3}$] |
|---|---|---|---|---|---|
| global | $4 \times 10^{-7}$ | 1.50-$i$0.0010 | 0.22 | 1.50 | 1.00 |
| 1st local | $3 \times 10^{-4}$ | 1.59-$i$0.0180 | 0.19 | 1.54 | 1.40 |
| 2nd local | $3 \times 10^{0}$ | 1.301-$i$0.000 | 0.80 | 1.35 | 0.12 |

(b)　in the 2nd case, there are 3 minima as well:

| Kind of Minimum | $F_g$ [%] | $m$ | $r_0$ [μm] | $\sigma$ | $n$ [cm$^{-3}$] |
|---|---|---|---|---|---|
| global | $3 \times 10^{-3}$ | 1.507-$i$0.0024 | 0.217 | 1.50 | 1.03 |
| 1st local | $5 \times 10^{-2}$ | 1.585-$i$0.0170 | 0.188 | 1.54 | 1.34 |
| 2nd local | $3 \times 10^{0}$ | 1.301-$i$0.0000 | 0.800 | 1.35 | 0.12 |

(c)　in the 3rd case, there are 2 minima:

| Kind of Minimum | $F_g$ [%] | $m$ | $r_0$ [μm] | $\sigma$ | $n$ [cm$^{-3}$] |
|---|---|---|---|---|---|
| global | $2 \times 10^{-1}$ | 1.540-$i$0.0100 | 0.21 | 1.51 | 1.11 |
| local | $3 \times 10^{0}$ | 1.301-$i$0.0000 | 0.800 | 1.35 | 0.12 |

We see that the global minimum coincides with the true solution only in the 1st case (a). The value of the global minimum is significantly lower compared to all other minimum values that we find in the three cases a–c. All elements of vector $\mathbf{x}^{\text{opt}}$ of the global minimum coincide with the respective elements of vector $\mathbf{x}^{\text{true}}$. The estimated accuracies of the $\mathbf{x}^{\text{opt}}$ elements as well as of all PMPs are of order of magnitude $\Delta_F$; see the retrieval results for the numerical example shown in Table 1.

In the other two cases, (b) and (c), the global minimum does not coincide with the true solution. Furthermore, we observe some smearing of the otherwise clearly located minima [see case a] for increasing uncertainty of the optical data. That means, whatever method of CRI retrieval we use, we will not be able to distinguish the different minima along the "canyon" which contains the global minimum (true solution) if we only use $3\beta + 2\alpha$ data. As we discussed we need extra constraints that allow us to localize the true solution.

The key question is*: does this "canyon" cover the true solution always, even if the optical data are distorted up to 10–15%?* As mentioned before we are investigating this question as the results will directly feed into the next version of TiARA and will have fundamental impact on the performance features of the current version 1.0 of TiARA [34].

Figure 7 shows 8 scenarios of extreme error distribution of the $3\beta + 2\alpha$ data we used in the numerical example. We use 15% data uncertainty for each data point; for details, we refer to [7]. The description of the error scenario can be found in the header of each case. We see that the "canyons" for each of the 8 extreme error cases always cover the true point $m = 1.5$-$i$0.001 (see white star). However, the global minima usually are located in different spots of these 8 canyons. We find these spots

-　At the beginning of the "canyon" in Figure 7a,c at point $m = 1.525$-$i$0.0005 ($F_g = 9.2\%$) and at point $m = 1.547$-$i$0.0 ($F_g = 8.1\%$), respectively. Both points are close to the true solution;
-　In the middle of the "canyon" in Figure 7g at point $m = 1.525$-$i$0.003 ($F_g = 2.9\%$);
-　At the end of the "canyon" in Figure 7b,d at points $m = 1.675$-$i$0.02 ($F_g = 3.7\%$) and $m = 1.65$-$i$0.03 ($F_g = 5.9\%$), respectively, and
-　Even at the beginning of the plane ($m_{\text{R}}$, $m_{\text{I}}$) in Figure 7e,f,h. The values of the complex refractive index in these cases are $m = 1.31$-$i$0.0022 ($F_g = 3.5\%$), $m = 1.34$-$i$0.001 ($F_g = 2.1\%$) and at $m = 1.325$-$i$0.0 ($F_g = 5.1\%$), respectively.

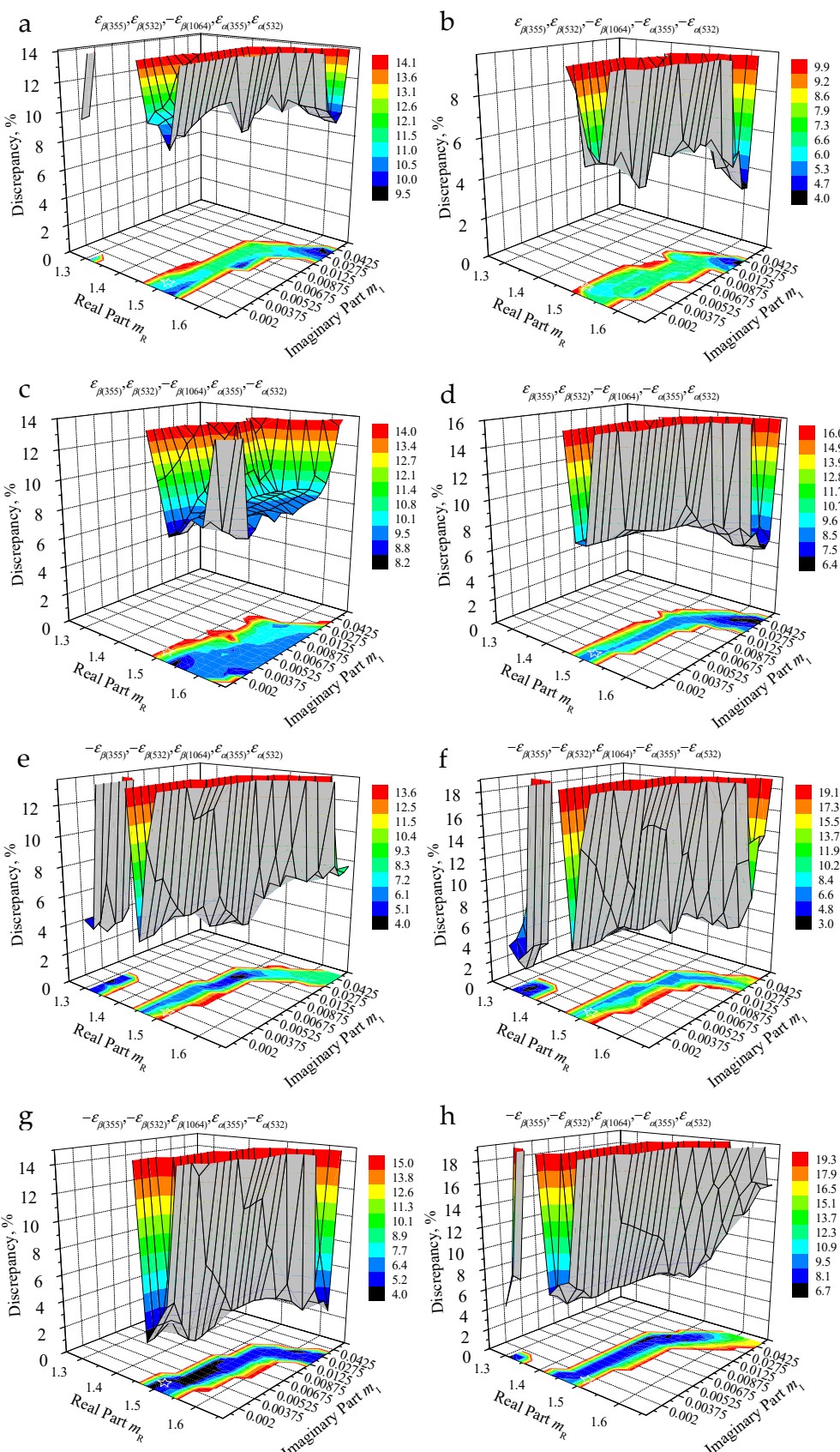

**Figure 7.** The same as Figure 6a but the optical data are distorted with extreme uncertainties of $\varepsilon_{g(\lambda)} = \pm15\%$. The distribution of the uncertainties between the five channels $3\beta + 2\alpha$ has been performed in different ways (**a**,**b**), ... and (**h**). The error scenario is shown in the header of each case (**a**–**h**).

The magnitude of the global minima $F_g{}^{\min}$ exceeds values of a few %. That level of more than just a few percent confirms that there is a high distortion of the optical data. Still, the "depths" of each "canyon" is approximately 10–20%, which agrees with the measurement errors of $\varepsilon_\alpha = 15\%$. Thus, the knowledge of the uncertainty of the optical input data is important. This knowledge allows us to estimate the width of the "canyon" for the different cases, which in turn allows us to completely cover the true CRI in our search algorithm.

This property of the "canyons"

- Holds true not only in the case considered in this example,
- It can be generalized for arbitrary combinations of elements of the vector **x**, and
- It is one of the achievements/advantages of state-of-the art multiwavelength lidar technologies [22,35] as we can find the CRI trajectory that contains the true solution.

However, we should keep in mind that there is no reason to reject any CRI value as potential/valid solution on this trajectory if the optical data uncertainty is 0.1–1% or more.

### 3.3. Use of Error Analysis

The results of the study carried out in the previous section shows that the CRI can be retrieved with a preset accuracy (up to $\Delta_F \sim 10^{-8}$) if the optical data $3\beta + 2\alpha$ are known up to 8 significant digits. In this case, any PMP ($r_0$, $\sigma$, $r_{\text{eff}}$, $n$, $s$ or $v$) is retrieved with similar accuracy $\sim \Delta_F$.

If the optical data are determined with an accuracy below 0.1%, which typically cannot be achieved with currently exiting lidar technology, the CRI uncertainty begins to appear. This uncertainty is described by the "canyon" that crosses the CRI domain from the lower left corner ($m_{\text{Rmin}}$, $m_{\text{Imin}}$) to the top right corner ($m_{\text{Rmax}}$, $m_{\text{Imax}}$) of the plane ($m_{\text{R}}$, $m_{\text{I}}$) shown in Figures 6 and 7. Apart from its length, the "canyon" is characterized by its width, which depends on the magnitude of the measurement error $\varepsilon_{g(\lambda)}$. In summary, we can state that $3\beta + 2\alpha$ lidar data and known measurement error provide us with the "canyon" that crosses the true CRI $m^{\text{true}}$. We find all other PMPs with PA in the "canyon".

Figure 8a shows the patterns of the retrieved parameters $r_{\text{eff}}$ (red), $n$ (green), $s$ (blue) and the product $nr_{\text{eff}}^2$ (yellow) versus $m$ in the "canyon" that is shown in Figure 6c. The optical data were determined with an accuracy of approximately 1%. We see that the effective radius decreases for increasing imaginary (real) part. Furthermore, an increase in the imaginary (real) part leads to an increase in number concentration. These patterns are usual, i.e., hold valid for any case of optical data and therefore they can be postulated.

**Table 2.** Aerosol types, respective CRIs at 532 nm, and growth factors (GF) for some levels of relative humidity (RH).

| Aerosol Types | RH < 5% | | $20\% \leq$ RH $\leq 25\%$ | | RH > 90% | |
|---|---|---|---|---|---|---|
| | GF | CRI | GF | CRI | GF | CRI |
| organic carbon | 1 | 1.53-$i$0.009 | 1.10 | 1.49-$i$0.007 | 2.5 | 1.35-$i$0.001 |
| black carbon | 1 | 1.75-$i$0.44 | 1 | 1.75-$i$0.44 | 1.9 | 1.43-$i$0.100 |
| sea salt | 1 | 1.50-$i$0.000 | 1.15 | 1.44-$i$0.000 | 2.5 | 1.35-$i$0.000 |
| sulfate | 1 | 1.43-$i$0.000 | 1.18 | 1.40-$i$0.000 | 2.0 | 1.35-$i$0.000 |
| dust | 1 | 1.53-$i$0.0026 | 1 | 1.53-$i$0.0026 | 1 | 1.53-$i$0.0026 |

However, in general, the overestimation of number concentration and underestimation of the effective radius can exceed $\sim$50% and $\sim$20%, respectively (see $m = 1.65$-$i$0.026). Conversely, underestimation of number concentration and overestimation of the effective radius occur at other corners of the "canyon" or the ($m_{\text{R}}$, $m_{\text{I}}$) plane (see $m = 1.301$-$i$0.0). Thus, the quality of the retrieved PMPs depends on the length of the "canyon", i.e., the domain $D$, and the position of the true CRI in the "canyon".

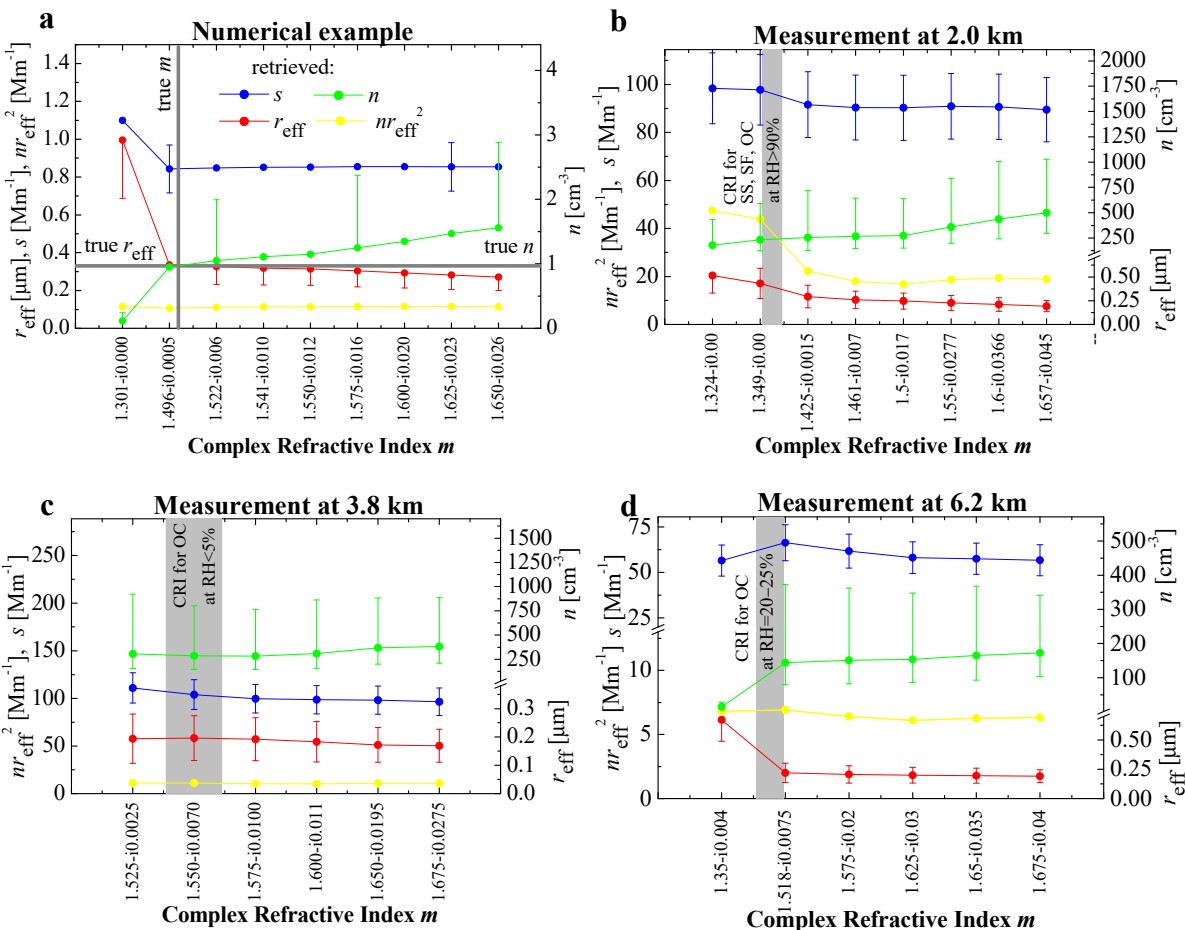

**Figure 8.** Results of the retrieved parameters $n$ (green), $r_{eff}$ (red), $s$ (blue) and $nr_{eff}^2$ (yellow) versus CRI in the "canyon" shown in Figure 6c for the numerical example (**a**) and in the "canyons" shown in Figure 6d,e,f for the lidar measurement from 22 July 2004 at heights 2.0 km (**b**), 3.8 km (**c**) and 6.2 km (**d**) [see details in Section 4], respectively. Uncertainty bars describe the retrieval uncertainties for 15% measurement uncertainty and the assumption that the single mode of the retrieved monomodal PSDs is positioned in the accumulation mode. (**a**) True CRI, the effective radius, surface-area and number concentrations are $m^{true}$ = 1.5-$i$0.001 (gray vertical line), $r_{eff}^{true}$ = 0.33 μm (gray horizontal line), $s^{true}$ = 0.85 Mm$^{-1}$ (not shown) and $n^{true}$ = 1 cm$^{-3}$ (gray horizontal line). The gray area describes the CRI of sea salt (SS), sulfate (SF) and/or organic carbon (OC) on the condition that relative humidity is more than 90% (**b**), less than 5% (**c**) and approximately 20–25% (**d**) [see Table 2].

We can use our error analysis for estimating the retrieval uncertainties of the PMPs. In the case of surface-area concentration, its uncertainty $\varepsilon_s$ is determined by Equation (19) [see also Figure 2a]. Since the magnitude of the uncertainty $\varepsilon_{\alpha(355)}$ may change from −15% to +15%, both overestimation and overestimation are possible (see blue error bars in Figure 8a). The $r_{eff}$, $v$ and $n$ uncertainties are determined from using Equations (24), (28) and (30) and by taking into account all CRI values which belong to the "canyon" on the ($m_R$, $m_I$) plane.

Figure 8a shows the uncertainties of $r_{eff}$ (red error bars) and $n$ (green error bars) for the numerical example considered in Figure 6c and the measurement uncertainty $\varepsilon_\alpha$ = 15%. The uncertainties depend on the perturbations of the EAE. The EAE in the numerical example is about $\dot{\alpha} \approx 0$. Therefore, its perturbed values may only lead to an overestimation of this level on the two conditions that measurement uncertainties and the accumulation mode fulfill the conditions of Equation (16) and the intervals in (15), respectively (see black line + circle, lower-right trajectory in Figure 2b].

As we discussed above, the overestimated value can reach $\dot{\alpha}^\varepsilon \approx 0.75$ for a measurement uncertainty of +15%. In turn, the overestimation of EAE results in underestimations of

$r_{eff}$ (see red error bars in Figure 8a) and $v$ (not shown). For the value of $m = 1.52\text{-}i0.006$ we can use the regression condition in Equation (22) in which $b_r/a_r = -0.32/0.12 = -2.67$ and Equation (24). Our estimate of the uncertainty of $r_{eff}$ is $\varepsilon_{r_{eff}} = 0.75/(0 - 2.67) = -28\%$. In view of Equations (28) and (30), we immediately find the uncertainties of $v$ and $n$ as $\varepsilon_v \approx \varepsilon_{r_{eff}} \approx -28\%$ and $\varepsilon_n = +93\%$ (see sub- and superscripts of the numerical example in Table 1).

The uncertainties can be estimated for all other CRI values along the "canyon" (see *x*-axis in Figure 8a). We conclude that a measurement error of 15% is considerably high, not only in the context of retrieving the CRI but also in view of retrieving all other PMPs. This magnitude of error leads to an uncertainty of the effective radius between 0.2 and 1.0 µm (red curve and uncertainty bar in Figure 8a) and between 0.1 and 3 cm$^{-3}$ for number concentration (green curve and uncertainty bar in Figure 8a).

We emphasize that the numerical example considered in our simulation is not specific. Results derived in the simulation can be generalized for any combination of CRI real and imaginary parts, mean radius and standard deviations. We are summarizing in the following list the major steps of our uncertainty analysis we use for any (arbitrary) case of optical/microphysical data and any PMP retrieval method:

1. Scan of the full domain of the CRI with the available method that is used to retrieve the solution space (see Figure 5).
2. Construction of the CRI solution trajectory by using the retrieved solution space (see Figures 6 and 7). This trajectory describes the "canyon" on the $(m_R, m_I)$ plane.
3. Identification of the final solution space by applying the minimization procedure to the CRI solution trajectory if the optical input data are acquired with an accuracy better than $\Delta_F \sim 10^{-8}$ (see Figure 6a).
4. Investigating the PMPs that are linked to the CRI solution trajectory and estimating their uncertainties that depend on the measurement uncertainty $\varepsilon_\alpha$ of the optical data (see Figure 8).
5. Decision making about the final solution space (Figure 8). This final decision is based on (mathematical and physical) a priori information as for example measurement uncertainties, aerosol typing, positivity (no negative number concentrations) of the retrieved PSDs, smoothness (modality) of the retrieved PSDs (see Section 3.1 and point 5 of Section 4).

## 4. Case Study

We carried out a first validation test of the capabilities of the minimization procedure and error analysis for the case of real measurement conditions. Multiwavelength lidar measurements have frequently been carried out in various regions worldwide [22,34–36]. A large amount of optical data has been acquired in the past two decades. Unfortunately, only very few experiments included simultaneous lidar and airborne in situ measurements, e.g., [37], which is a necessary condition for trustworthy and robust validation tests.

We investigate a measurement from 22 July 2004 [38]. In that case, in situ data are available, though this case does not fulfill the ideal conditions in a strict sense as lidar observations and in situ observations were not collocated (in space and in time). The top of the planetary boundary layer reached 2.5 km during the time of the measurements. Forest-fire smoke plumes were encountered between 4 and 5 km altitude. We considered three aerosol layers in our case study. These layers were located at heights of 2.0 km, 3.8 km and 6.2 km above sea level.

If we apply the results of our methodology to the particle layer at height 2 km, we find the following results:

1. We retrieve the solution space by applying PA to the optical dataset $3\beta + 2\alpha$. Figure 5b shows the PSDs and respective PMPs (see legend). The number concentration in this solution space varies between 214 and 728 cm$^{-3}$, the mean radius varies between 0.115 and 0.075 µm, the CRI varies between $1.35\text{-}i0.0$ and $1.7\text{-}i0.05$, and SSA at 532 nm varies between 0.81 and 1.00 (see solutions in gray and blue). The retrieved PSDs can

be both monomodal and bimodal. The total effective radius is 0.16 μm in the case of a monomodal PSD (gray) whereas it increases to 0.35 μm in the case of a bimodal PSD (red).

2.  We construct the CRI trajectory on the ($m_R$, $m_I$) plane from the retrieved solution space. The CRI trajectory is extended from $m = 1.3\text{-}i0.0$ to $1.7\text{-}i0.05$ (see Figure 6d) for all solutions with discrepancy $\rho \leq 10\%$ [3,10,33].

3.  We use the minimization procedure to find the minima in the "canyon". We find three minima (see Figure 6d):

| Kind of Minimum | $F_g$ [%] | $m$ | $r_0$ [μm] | $\sigma$ | $n$ [cm$^{-3}$] |
| --- | --- | --- | --- | --- | --- |
| global | 0.01 | 1.657-$i$0.045 | 0.862 | 1.77 | 501 |
| 1st local | 2.0 | 1.461-$i$0.007 | 0.120 | 1.74 | 269 |
| 2nd local | 4.0 | 1.324-$i$0.000 | 0.115 | 2.17 | 178 |

4.  We also find the PMPs in the "canyon" and estimate their uncertainties depending on the measurement uncertainty of the optical data, i.e., $\varepsilon_\alpha = 15\%$. Figure 8b shows the retrieved surface-area (blue) and number (green) concentrations, the effective radius (red) and respective uncertainty bars versus CRI and the assumption that the PSDs are monomodal. Number concentration varies from 125 to 1030 cm$^{-3}$, surface-area concentration from 77 to 113 Mm$^{-1}$, the effective radius from 0.13 to 0.6 μm, the real part of the CRI from 1.324 to 1.657 and the imaginary part from $i$0.0 to $i$0.045.

5.  The solution space we identified in Figure 8b, and assuming a measurement uncertainty of 15%, shows a widespread of values for the PMPs. We investigate if this spread can be reduced.

As we have previously shown, four parameters, such as number concentration, the effective radius, real and imaginary parts of the CRI are mutually dependent. Figure 8 particularly shows that increasing values of the real and imaginary parts of the CRI ($x$-axis) result in decreasing values of the effective radius (red curve). This interdependency means that if we know one of the four parameters we can find the values of the other three parameters. We illustrate this property in Figure 8a by the horizontal and vertical gray lines which cross each other at the true values, i.e., $m^{\text{true}} = 1.5\text{-}i0.001$, $r_{\text{eff}}^{\text{true}} = 0.33$ μm and $n^{\text{true}} = 1$ cm$^{-3}$.

This interdependency that we observe for $m$ and $r_{\text{eff}}$ resembles features of the size growth of hygroscopic particles in which mean particle size and the complex refractive index vary with ambient atmospheric RH [14]. The strong resemblance of this pattern of interdependence triggered us to test (in terms of a strong hypothesis) if relative humidity data can be of value in our search for the final solution space.

In fact, hygroscopicity is an important feature of aerosol particles in general. It is not only another data product provided by lidar. More importantly it (1) is a standard data product acquired in many in situ aerosol observations under field and laboratory conditions, (2) can be used for more accurate aerosol typing, and (3) plays a vital role in any kind of aerosol characterization in general [14,39,40].

Most Raman lidar instruments currently in use in, e.g., ACTRIS/EARLINET either measure relative humidity or can easily be upgraded to deliver this data product. Thus, it is reasonable to explore what extra constraints in data inversion, respectively solution space identification, are possible if we hypothesize that particles are hygroscopic. Here, we use this information for illustration as it would provide us with substantial added value for our methodology, i.e., if precise data on hygroscopic growth for complex mixtures of aerosols and pure aerosol types were regularly provided.

We are aware of the complexity of studies of particle hygroscopic growth, and the knowledge needed to separate various effects connected to, e.g., instrument performance effects before any robust information on particle hygroscopic growth can be acquired. Our study presented by Sawamura et al. [37] provided us with detailed insight on the challenges involved in such studies. We stress therefore that this test is merely a first attempt of a more detailed study and by no means is intended to lead to general conclusions in the context of this paper.

Table 2 contains some data of relative humidity and respective growth factors, and CRIs for some of the most widespread atmospheric aerosol types. We selected data from [21] which provides us with information on dust (D), black carbon (BC), organic carbon (OC), sea salt (SS) and sulfate (SF) particles of MERRA-2 model. For example, values of CRI for organic carbon in this specific publication show a decrease from $m = 1.53$-$i0.009$ to $1.35$-$i0.001$ at $\lambda = 532$ nm and an increase in the growth factor from 1 to 2.5 for the case that RH increases from 0% to 100%.

Measurements of the optical data profiles from 22 July 2004 included simultaneous observations of RH with radiosonde [38]. Relative humidity measured with radiosonde at 2 km is more than 90%. We select from Table 2 the CRIs $m = 1.35$-$i0.0$ for SS and SF particles as well as $m = 1.35$-$i0.001$ for OC particles at RH > 90% because for these aerosol types the values $m = 1.35$-$i0.0 \ldots 0.001$ are intermediate between $m = 1.349$-$i0.0$ and $m = 1.425$-$i0.0015$ on the $x$-axis of Figure 8b. We insert the respective CRI values into Figure 8b (see grey area). As we can see, all other CRIs from Table 2 at RH > 90% (BC and D) do not match with the $x$-axis nodes in this case.

There is a pronounced global minimum at $m = 1.657$-$i0.045$. The discrepancy is $\rho^{\min} = 0.01\%$, see Figure 6d. As we discussed before, the challenges related to the CRI estimation cannot be overcome by a simple choice of the available solution methods of Equation (1). We have to search for extra constraints that may allow us to localize the true CRI on the solution trajectory. Under the assumption that the observed aerosols were hygroscopic we used the values $m = 1.35$-$i0.0 \ldots 0.001$ (grey area) as constraint for the CRI solution space. We included in the final solution space the CRI solution $m = 1.349$-$i0.0$ with discrepancy $\rho^{\min} = 4\%$. The solutions $m = 1.349$-$i0.0$ and $m = 1.657$-$i0.045$, which are on the opposite corners of the $(m_R, m_I)$ plane show why, for instance, our method of inversion with regularization [2,7] tends to overestimate the CRI (see also [38]). The overestimation is the result of very specific properties of the solution space and not the uncertainty caused by the method used for identifying the solution space.

We also use PA for the case of the bimodal PSDs which are obtained at 2 km height (see red and green solutions in Figure 5b). Bimodal PSDs retrieved with PA produce discrepancies $\rho \leq 0.5\%$. For that reason, we have to include these solutions, too, in the final solution space that contains the monomodal PSDs; see, e.g., the PSD in blue for which the CRI is approximately $1.35$-$i0$. However, we can remove the red bimodal PSD from the final solution space because the coarse mode shows an unrealistically low value of the real part of the CRI of $m_R = 1.3$.

We can now make the final decision on the solution space, which is driven by the task of collecting the monomodal (blue) and bimodal (green) PSDs. Despite the fact that these PSDs are quite different of each other we can restrict the spread of the PMPs, i.e.,

$$
\begin{aligned}
r_{\text{eff}} &\in [0.31;0.43] \ \mu\text{m}; \quad v \in [8.3;13.8] \ \mu\text{m}^3\text{cm}^{-3}; \\
n &\in [147;277] \ \text{cm}^{-3}; \quad \text{SSA}(532) \in [0.95;1.00].
\end{aligned}
\tag{36}
$$

The solutions in these intervals result in the following values for the PMPs: $r_{\text{eff}} = 0.37(\pm16\%) \ \mu\text{m}$, $v = 11(\pm25\%) \ \mu\text{m}^3\text{cm}^{-3}$, $n = 212(\pm30\%) \ \text{cm}^{-3}$, $\text{SSA}(532) = 0.975 \pm 0.025$. The quality of the measurement data at 2 km may be high since the discrepancies of the retrieval results are only a few percent (see Table 1). Therefore, these solutions could be taken as the final result for the solution space. The uncertainty bars for $r_{\text{eff}}$, $v$ and $n$ (see uncertainty bars in Figure 8b and sub- and superscripts in Table 1) can be estimated for 15% measurement uncertainty from Equations (19), (24), (28) and (30).

However, we have limited knowledge on the measurement conditions and the instrument performance. Thus, the measurement uncertainty of the optical data plays a crucial factor in our case study. We also must keep in mind the unfavorable situation that the in situ observations we used for our comparison did not coincide with the location of the lidar observations.

We analyzed the measurement data at heights 3.8 and 6.2 km as well. The results are shown in Figures 5c, 6e, 8c, 5d, 6f and 8d, respectively. The solution spaces of both

measurement cases show a widespread of the PMPs. Number concentrations range from 18 to 455 cm$^{-3}$. Mean radii vary from 0.435 to 0.115 μm (see blue curve in Figure 5d and gray curve in Figure 5c). Again, we find monomodal and bimodal PSDs solutions. We restrict the spread of the retrieved PMPs by constraining the range of values of the CRI by using information about relative humidity at 3.8 and 6.2 km.

Relative humidity measured with radiosonde at 3.8 km is less than 5%. We select from Table 2 the CRI $m = 1.53\text{-}i0.009$ for organic carbon at RH < 5% because for this aerosol type the value $m = 1.53\text{-}i0.009$ is intermediate between $m = 1.525\text{-}i0.0025$ and $m = 1.575\text{-}i0.01$; see the *x*-axis of Figure 8c. We use this value for the CRI in Figure 8c. This value acts as constraint on the solution space of the CRI (see grey area). We see that all other CRIs in Table 2 do not match the x-axis nodes at RH < 5%; see the "canyon" in Figure 6e and the *x*-axis in Figure 8c. If we use the PMPs that result from the retrieved monomodal PSDs at the CRI in the proximity of $m = 1.53\text{-}i0.009$ and also from the bimodal PSDs, we can decide on the final solution space. We find

$$r_{\text{eff}} \in [0.20;0.45] \text{ μm}; \quad v \in [6.8;13.7] \text{ μm}^3\text{cm}^{-3}; \\ n \in [286;372] \text{ cm}^{-3}; \quad \text{SSA(532)} \in [0.96;0.99]. \tag{37}$$

The solution discrepancies of the monomodal PSDs for this case exceed 11% which means that measurement errors may be at least 15% or even higher. Therefore, optical data must always be provided with measurement uncertainties as that allows us to provide uncertainty bars for the microphysical parameters. 15% error results in the uncertainty bars estimated from Equations (19), (24), (28) and (30) and shown in Figure 8c and in Table 1 (see sub- and superscripts in %). Still, we believe that SSA can be estimated to $0.975 \pm 0.015$ despite comparably high measurement uncertainties if we have information on relative humidity. In the ideal case, we prefer of course information on the hygroscopic growth of particles that belong to specific aerosol types.

Relative humidity measured with radiosonde at 6.2 km is about 20%. We again select the value $m = 1.49\text{-}i0.007$ for OC from Table 2 as constraint for this measurement case. Only that value (Table 2) at RH = 20–25% matches the trajectory of the solution of the CRI (see "canyon" in Figure 6f and *x*-axis in Figure 8d). The final result of the PMPs in that case is

$$r_{\text{eff}} \in [0.22;0.46] \text{ μm}; \quad v \in [4.9;8.0] \text{ μm}^3\text{cm}^{-3}; \\ n \in [130;167] \text{ cm}^{-3}; \quad \text{SSA(532)} \in [0.94;0.96]. \tag{38}$$

The solution discrepancies of the monomodal PSDs are 15–26%. There may be various reasons for this result. We find the highest measurement uncertainties at the top part of the vertical profile of the optical data, but we also find a multimodality of the PSDs. Although we usually consider bimodal PSDs as the main test case for our exploratory studies, it is clear from many observational studies that at least three to four modes are needed to describe atmospheric particle size distributions. In fact, the discrepancies of the retrieved bimodal PSDs are less than 2%.

If the true PSD is bimodal, we can still use the retrieval uncertainties we obtain for the PMP from applying Equations (19), (24), (28) and (30), see Figure 8d (uncertainty bar) and Table 1 (sub- and superscripts in %). We note that the SSAs of the bimodal and monomodal PSDs of the final solution space differ by 0.02 (at 532 nm), which means the SSA retrieval uncertainty is $\varepsilon_{\text{SSA}} = \pm 0.01$.

Figure 8 also shows the data product $nr_{\text{eff}}^2$ (yellow). This parameter is almost constant with regard to changes in the CRI in the numerical example (a), and in the measurement cases at 3.8 km (c) and 6.2 km (d). This constant interdependency means that the retrieval uncertainty of number concentration is determined by the square of the retrieval uncertainty of the effective radius if we use Equation (30).

The measurement case at 2.0 km (see Figure 8b) shows a sudden change in the product $nr_{\text{eff}}^2$ for the complex refractive indices of $m = 1.324\text{-}i0$ and $m = 1.349\text{-}i0$. This change is explained by the fact that the PSDs at these CRIs show a variance of $\sigma \approx 2.15$(see blue curve

in Figure 5b). At the same time, we find variances of $\sigma$ = 1.75–1.85 for PSDs retrieved at the other CRIs (see gray, black and pink curves in Figure 5b). That means there exists a significant shift in the properties of the PSDs in these different CRI subdomains. If we want to keep the product $nr_{\mathrm{eff}}^2$ constant, the correlation condition (29) needs to be corrected for with a factor that depends on variance. For PSDs described by logarithmic-normal distributions (7), one can show that the following strict equality holds true

$$s \equiv 4\,\pi\,n\,r_{\mathrm{eff}}^2\,\exp(-3\ln^2\sigma). \tag{39}$$

In conclusion, we showed for this case study that the retrieval results of the PMPs as well as the retrieval uncertainties of the PMPs agree if we either apply the new uncertainty analysis scheme or the two-dimensional regularization approach [38]. Our results and the in situ data available at height 3.8 km agree, too, for the effective radius, number, surface-area and volume concentrations. We find significant differences for SSA. Our analysis results in a value of $0.975 \pm 0.015$ whereas the in situ data show $0.91 \pm 0.03$. This lower value is linked to an imaginary part of the CRI of approximately $i0.02$. The trajectory of the solutions of the CRI inferred with our uncertainty analysis contains the value $m = 1.55\text{-}i0.02$ (see star in Figure 6e). However, we do not consider $m = 1.55\text{-}i0.02$ in our final solution space because this value of the CRI neither matches the model we use (see Table 2) nor what we found in [21].

We stress, that the approach of constraining the CRI is very exploratory. We need more studies to validate this methodology from the theoretical as well as from the experimental points of view, respectively. Further, we must keep in mind:

- Lidar measurements are not accurate and measurement uncertainty can exceed the level of 15% we used in our error analysis in this study;
- Radiosonde measurements also have uncertainties, which means that constraints we use to localize the true CRI on solution trajectories are not correct;
- Particularly, with regard to radiosonde, we must keep in mind that the balloons usually do not measure the same air masses as the lidar system does because balloons (a) either drift away from the lidar beam during ascent or (b) lidar site and balloon site are not collocated at all;
- Finally, the model MERRA-2 that we use is not ideal and may not properly describe the particle hygroscopic features. There are many different hygroscopicity observations published in literature and these results may likely lead to a spread of the correlations shown here.

In view of all these reasons, we keep the bimodal PSDs in the final solution spaces, which in turn influences the values we obtain for SSA. If we consider the SSAs that follow from the mono- as well as the bimodal PSDs, we obtain SSA retrieval uncertainties of $\varepsilon_{\mathrm{SSA}} = \pm 0.01 \ldots 0.025$ [see solutions in (35)–(37) and Table 1].

## 5. Conclusions

We developed a minimization procedure that allows for the simultaneous retrieval of particle microphysical parameters (PMP) such as number, surface-area and volume concentrations, the effective radius, and the complex refractive index from optical data ($3\beta + 2\alpha$) taken with multiwavelength HSRL/Raman lidar. The minimization procedure allows us to estimate the PMPs and the CRIs with any preset accuracy, if the optical data are known to eight significant digits. This condition of course is an unrealistic assumption in the case of experimental data. However, by starting with this assumption we gained some fundamental insight on the properties of the retrieved solution spaces, which allowed us to subsequently develop this minimization procedure as well as a new error analysis method.

The method of uncertainty analysis is based on estimating the PMP retrieval uncertainties $\varepsilon_p$ in dependence of the magnitude of the measurement uncertainties of the extinction coefficients $\varepsilon_{\alpha(355)}$ and $\varepsilon_{\alpha(532)}$ (see steps 1–5 in Section 3.3). We investigated the uncertainties of PMP retrievals in dependence of realistic measurement uncertainties of $3\beta + 2\alpha$ optical

data by using correlation relationships that we found in previous studies [19,25]. We show that

- The retrieval uncertainty of the CRI can be described by a trajectory ("canyon") that crosses the complete (search) domain of the CRI ($m_R$, $m_I$) from its lower-left corner, which means $m_{Rmin}$, $m_{Imin}$, to the top-right corner, which means $m_{Rmax}$, $m_{Imax}$. The "canyon" length and width are determined by the CRI domain and measurement uncertainty, respectively. The retrieval uncertainty $\varepsilon_s$ of surface-area concentration is proportional to the measurement uncertainty $\varepsilon_{\alpha(355)}$ of the extinction coefficient at 355 nm [see Equation (19)].
- The retrieval uncertainty of the effective radius ($\varepsilon_{reff}$) is inversely proportional to the uncertainty of the measurement uncertainties of the extinction-related Ångström exponent (EAE). This uncertainty is determined by the uncertainties of the measured extinction coefficients [$\varepsilon_{\alpha(355)}$ and $\varepsilon_{\alpha(532)}$]. We find uncertainties $\varepsilon_{reff}$ that are 1.8 ... 16.7 times larger than the measurement uncertainties $\varepsilon_{\alpha(355)}$ [see Equations (24) and (25)].
- The retrieval uncertainty $\varepsilon_v$ of volume concentration is close to $\varepsilon_{reff}$.
- The retrieval uncertainty $\varepsilon_n$ of number concentration is proportional to the inverse square of the retrieval uncertainty $\varepsilon_{reff}$ of the effective radius [see Equation (30)].

The retrieval uncertainties $\varepsilon_s$ and $\varepsilon_v$ can be estimated for arbitrary (multimodal) particle size distributions (PSD), too. The retrieval uncertainties $\varepsilon_{reff}$ and $\varepsilon_n$ are valid for arbitrary monomodal PSDs. These PSDs contain the optically most-active particles, i.e., particles in the accumulation mode. The retrieval uncertainty may increase, for example, for bimodal PSDs. Therefore, we also consider the retrieval results for bimodal PSDs, which allows us to keep all possible (mono- and bimodal) cases in the final solution space.

We show on the basis of a case study that measurement uncertainties of $\varepsilon_{g(\lambda)} = 15\%$ of $3\beta + 2\alpha$ optical data result in a considerable spread of the solution space, not only with regard to the CRI but also with regard to the other PMPs. Usually, these solutions consist of monomodal and bimodal particle size distributions. No matter what retrieval method we use, we cannot suppress this spread of the solutions without applying constraints on the solution space. In the first step, we used information about relative humidity as extra constraint, which is a realistic approach. Most Raman lidar instruments currently in use in, e.g., EARLINET either measure relative humidity or can easily be upgraded to deliver this data product.

We emphasize that the purpose of using hygroscopic growth information under varying relative humidity conditions (as shown in this publication) was to explore if such information could further constrain the solution space. We neglected the fact that there are extreme challenges involved in determining the hygroscopic growth of particle types and mixtures of aerosol types to sufficient precision. However, we believe that our study points to a new direction in determining microphysical particle properties from the inversion of lidar data that contain relative humidity information. The estimation of accurate values of the particle microphysical parameters, including real and imaginary parts of complex refractive indexes, could lead to the retrieval of the particle chemical properties aloft, as shown in [41,42].

Neglecting the challenges involved in characterizing the complex nature of particle hygroscopic growth we find that the constraints that follow from using relative humidity as additional information allow us to retrieve single scattering albedo (at 532 nm) with an uncertainty that is no larger than $\varepsilon_{SSA} = \pm 0.01 \ldots 0.025$. This finding rests upon a limited set of lidar data for which we could compare the retrieval results to in situ data on the condition that the measurement uncertainties of the optical data were less than $\varepsilon_{g(\lambda)} \leq 15\%$. We will continue with this work and our exploratory tests of this novel idea. Data are available from other field campaigns in which lidar and in situ data of aerosol particles were taken in a near simultaneous fashion, e.g., DISCOVER-AQ and ORACLES [22,41]. All the results derived in our study will be used in future uncertainty analysis of data products provided by TiARA.

**Author Contributions:** Conceptualization, methodology, writing—original draft preparation, A.K.; writing—review and editing, funding acquisition, D.M.; software, A.R. All authors have read and agreed to the published version of the manuscript.

**Funding:** This research was funded by Russian Science Foundation (project 21-17-00114).

**Data Availability Statement:** The data presented in this study are available in [38].

**Conflicts of Interest:** The authors declare no conflict of interest.

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
