# Peer review of "Particle Microphysical Parameters and the Complex Refractive Index from 3β + 2α HSRL/Raman Lidar Measurements: Conditions of Accurate Retrieval, Retrieval Uncertainties and Constraints to Suppress the Uncertainties"

_atmosphere, doi:10.3390/atmos14071159_

Round 1

Reviewer 1 Report

The article discusses the potential of High Spectral Resolution Lidar (HSRL) and Raman Lidar measurements for accurately retrieving particle microphysical parameters (PMP) such as number, surface-area, and volume concentrations, effective radius, and complex refractive index of atmospheric particles. The authors analyze the uncertainties of PMP retrievals due to measurement errors in 3beta+2alpha optical data and present results that are important for understanding how uncertainties of optical data convert into uncertainties of PMP. They found that accurate retrieval of PMP requires optical data with at least 8 significant digits accuracy, which is not achievable with currently existing lidar measurement techniques. The study also shows that constraints such as relative humidity can reduce retrieval uncertainty of single scattering albedo to as low as +/-0.01 - +/-0.025 (at 532 nm). The results will be used for uncertainty analysis of data products provided by future versions of TiARA. On the other hand, the uncertainty of effective radius is inversely proportional to the measurement uncertainty of the extinction-related Ångström exponent. The study also found that complex refractive index cannot be estimated without introducing extra constraints, even if measurement uncertainties of the optical data are as low as 1-3%. Overall, the article highlights the potential of HSRL/Raman lidar measurements for studying aerosol properties and their effects on climate while also emphasizing the need for accurate optical data to achieve precise retrievals of PMPs. The article has well-organized data and clear technical procedures. Therefore, I recommend it for publication.

Recommendation: Please rewrite the equations(33)(36)(38). Alternatively, these equations should be included in a table.

Reviewer 2 Report

his article deal with the method to retrieve aerosol CRI using 3beta-2alpha. 

As authoe have said, this method have fundamental limitation because of detection  error and methematical peroperties in the solution space. To overcom these limitation they have suggested some constraints on the solution from priori values, RH, and theoretial backgrounds.

This article can be publisged in this article, But I want to suggest to rearragge some section and  add some sentences for the reader to understand more easily.

1)

THis article deal with CRI information from 3b+2a data using "priori informations" and  specific size distribution such as "nono-lognormal".

THis conditions will be satisfied in high altitude aerosol, so this article title should be changed to increase the readibility.

I hope to change the title  include some information "priori informations" or "nono-lognormal"

for example:

Particle Microphysical Parameters and Complex Refractive In-2 dex from 3b+2a HSRL/Raman Lidar Measurements: Retrieval  Uncertainties Versus Measurement Errors using constraints and priori values

or  

For example, Please insert some sentences in the "introduction section" such like as line 480-492.

2) I think section 2-3 is bulky and unnecessary section

if author want to include section 2-3 they should include retrieval characteristics which depend only on the CRI.

2) There are too much additional referece and ncessary figures for explaining results ( for example line 771 : reader should read reference [36] and figure in it  )

-for each figure please explain the discussion at the end of figure, not far from the figure.

3)  About SSA

For example, SSA is not independent parameter, it can be calculated from  PMPs. so we donot need SSA in Eq(36), 37, 38 to understand CRI retrieval process.

On the other hand I cannot find any retrieval peoperties about variance=sigma( one of the four independent parameter)

Instead of SSA, the author should include sigma in Eq. 36 , 37, 38.

If variance is not important independent parameter, please explain the reason why aerosol variance is not important in the  measurements valuse G or g.

4) PMPs are 4 parameters, ro, sigma, m=n+ik

 I cannot find any information about sigma, As i know sigma have little contribution to the 4 -measurement vaules (G={ G_j }=EAE, Lidar ratios, BAE).

How abou to fix-sigma value at the middle point of [1.35;2.55]?

Question:

Figure 1: please discribe How can you increase  alpha(355). for example, by changing size(ro) in the Equation (17)

line 107:  CRI identified in point B; what doed 'B' means?

suggestion:

abstract : 8 digital bit -->10^-6  %

mistypings

line 427 : discrepancey (31-->32)

line 511:  convex structure. --> concave?

line 729: 1)--> (1)

Table 1 :v, um^2 cm^-3 -->um^3 cm&-3

As a person who speaks English as a foreign language, this article is difficult to read and has some awkward parts.

Reviewer 3 Report

see pdf-file

Round 2

Reviewer 3 Report

All my questions were answered.
